# Measurements of damage and repair of binary health attributes in aging mice and humans reveal that robustness and resilience decrease with age, operate over broad timescales, and are affected differently by interventions

Spencer Farrell[1]*, Alice E Kane[2], Elise Bisset[3], Susan E Howlett[3,4], Andrew D Rutenberg[5]*

[1]Department of Physics, University of Toronto, Toronto, Canada; [2]Blavatnik Institute, Department of Genetics, Paul F. Glenn Center for Biology of Aging Research at Harvard Medical School, Boston, United States; [3]Department of Pharmacology, Dalhousie University, Halifax, Canada; [4]Department of Medicine (GeriatricMedicine), Dalhousie University, Halifax, Canada; [5]Department of Physics and Atmospheric Science, Dalhousie University, Halifax, Canada

**Abstract** As an organism ages, its health-state is determined by a balance between the processes of damage and repair. Measuring these processes requires longitudinal data. We extract damage and repair transition rates from repeated observations of binary health attributes in mice and humans to explore robustness and resilience, which respectively represent resisting or recovering from damage. We assess differences in robustness and resilience using changes in damage rates and repair rates of binary health attributes. We find a conserved decline with age in robustness and resilience in mice and humans, implying that both contribute to worsening aging health – as assessed by the frailty index (FI). A decline in robustness, however, has a greater effect than a decline in resilience on the accelerated increase of the FI with age, and a greater association with reduced survival. We also find that deficits are damaged and repaired over a wide range of timescales ranging from the shortest measurement scales toward organismal lifetime timescales. We explore the effect of systemic interventions that have been shown to improve health, including the angiotensin-converting enzyme inhibitor enalapril and voluntary exercise for mice. We have also explored the correlations with household wealth for humans. We find that these interventions and factors affect both damage and repair rates, and hence robustness and resilience, in age and sex-dependent manners.

*For correspondence:
spencer.farrell@utoronto.ca (SF);
adr@dal.ca (ADR)

**Competing interest:** The authors declare that no competing interests exist.

## Editor's evaluation

The key contribution of this study is to evaluate the longitudinal change in frailty indices by tracking both accumulation of damage and repair of deficits (damage and repair transition rates), using a sophisticated mathematical modeling and a translational approach that spans mice and humans. A second key achievement of this study is to evaluate change in frailty indices and damage and repair transition in interventions that improve health in mice. Collectively this advances progress in translational geroscience by providing new insight regarding how we measure biological age that can

aid assessment of aging-relevant interventions. The authors have provided extensive details that support the research frameworks presented in this report.

## Introduction

As organisms age, they can be described by health states that evolve according to dynamical processes of damage and repair. A health state is the net result of accumulated damage and subsequent repair (*Howlett and Rockwood, 2013*). Studies of aging have mostly focused on discrete health-states rather than the underlying continuous dynamic processes, due the difficulty of their measurement. Two common approaches to measuring individual health-states, the Frailty Index (FI) (*Mitnitski et al., 2001*) and the Frailty Phenotype (*Fried et al., 2001*), are assembled from health state data at a specific age and do not separate dynamic damage and repair processes. Nevertheless, strong associations between frailty measures and adverse health outcomes (*Hoogendijk et al., 2019*; *Howlett et al., 2021*) indicate that frailty affects the underlying dynamical processes. This is supported by the increasing rate of net accumulation of health deficits with worsening health (*Mitnitski et al., 2007*; *Kojima et al., 2019*).

Reduced resilience, or the decreasing ability to repair damage (or recover from stressors), is increasingly seen as a key manifestation of organismal aging (*Ukraintseva et al., 2021*; *Kirkland et al., 2016*; *Hadley et al., 2017*). Resilience is often assessed by the ability to repair following an acute stressor, such as a heat/cold shock, viral infection, or anesthesia; or a non-specific stressor such as a change of the health state, typically within a short timeframe (*Scheffer et al., 2018*; *Gijzel et al., 2019*; *Rector et al., 2021*; *Colón-Emeric et al., 2020*; *Pyrkov et al., 2021*). Robustness, or an organism's resistance to damage, has not been as well studied – but there is also evidence for its average decline with age (*Arbeev et al., 2019*; *Kriete, 2013*). Both resilience and robustness sustain organismal health during aging, but their relative importance and their timescales of action remain largely unexplored.

While cellular and molecular damage and dysfunction are classic 'hallmarks' of aging (*López-Otín et al., 2013*), damage and dysfunction at organismal scales may exhibit distinct behavior (*Gems and de Magalhães, 2021*; *Howlett and Rockwood, 2013*). Indeed, from a complex systems perspective we may expect qualitatively distinct emergent phenomena at tissue or organismal scales (*Cohen et al., 2022*). However, a significant amount of organismal health data is discrete and cannot be approached with existing techniques used to study resilience or robustness. It is important both to study resilience and robustness at organismal scales and to be able to use discrete data while doing so.

To simultaneously study both resilience and robustness during aging with binarized health-deficits, we have here developed a novel method of analysis that uses longitudinal data from mice and humans to obtain summary measures of organismal damage and repair processes over time. This approach can be adapted to use any discrete biomarker. We apply our method to study how resilience and robustness evolve with age and how they differ between species, between sexes, and under different health interventions.

Our approach and results are limited to binarized health attributes; for our purposes damage and repair correspond to discrete transitions of these binarized attributes. Since our attributes are at the clinical or organismal scale of health, we do not consider cellular or molecular damage directly. Although our approach could be applied to binarized attributes at any organismal scale, we do not investigate whether our conclusions generalize to different sets of binarized attributes, nor do we consider continuous attributes.

Developing interventions to extend lifespan and healthspan is the goal of geroscience (*Kennedy et al., 2014*; *Sierra, 2016*; *Sierra et al., 2021*). While some interventions that affect aging health have been identified, how they differentially affect damage and repair, and their timescales of action, is less understood. We consider interventions in mice that have previously been shown to have a positive impact on frailty, the angiotensin converting enzyme (ACE) inhibitor enalapril (*Keller et al., 2019*) and voluntary exercise (*Bisset et al., 2022*). In humans, we stratify individuals within the English Longitudinal Study of Aging by net household wealth (*Phelps et al., 2020*; *Steptoe et al., 2014*). Wealth is a socioeconomic factor associated with aging health (*Zimmer et al., 2021*; *Niederstrasser et al., 2019*). Understanding how various interventions affect aging health by affecting resilience and robustness will better enable us to fulfill the geroscience agenda.

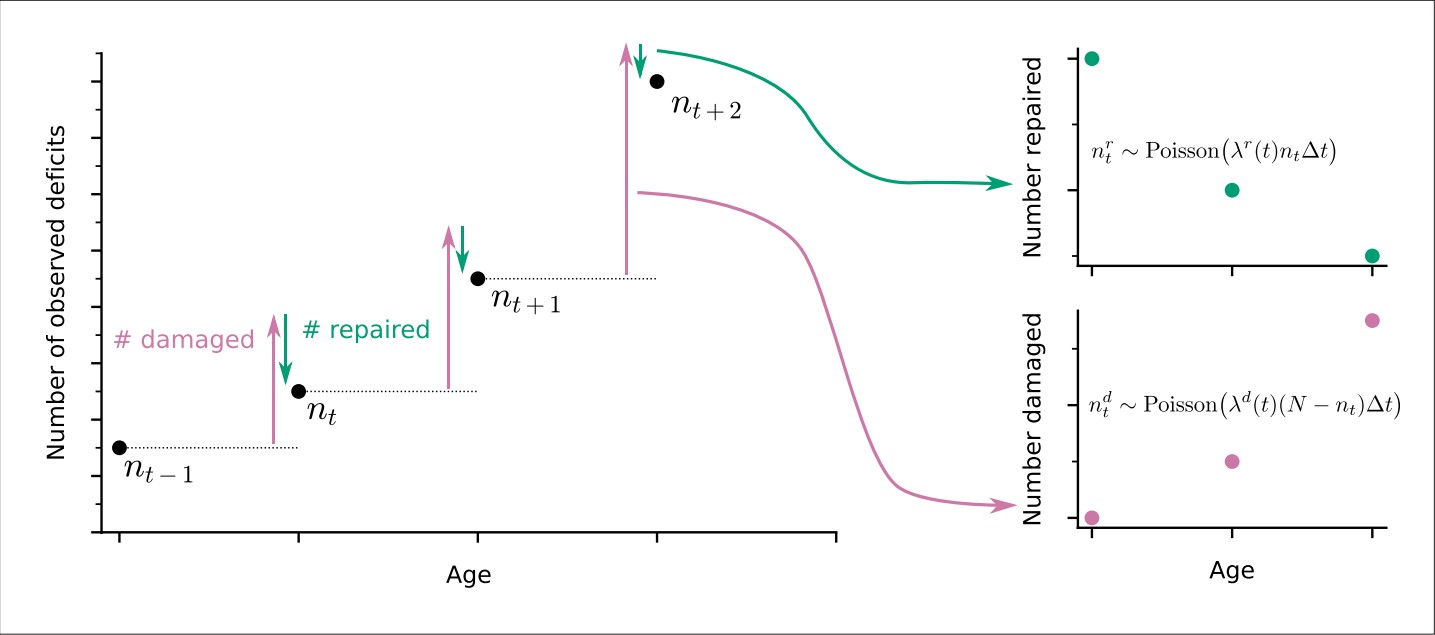

**Figure 1.** Extracting damage and repair from the longitudinal observation of binary health deficits. Instead of just considering the Frailty Index (FI) or net count of deficits at each age $n_t$ (i.e. FI multiplied by the total number N of deficits considered) as a measure of health, we separately consider the number of deficits damaged $n_t^d$ or repaired $n_t^r$ within a time interval $\Delta t$. Time-dependent damage $\lambda^d(t)$ and repair $\lambda^r(t)$ rates are extracted using Poisson models for the counts of repaired or damaged deficits.

## Results

### Measuring resilience and robustness with binarized data

A well-established approach to quantify health in both humans and in animal models is to count binarized health deficits in an FI (*Mitnitski et al., 2001*; *Whitehead et al., 2014*). In longitudinal studies, the FI can be assessed at each follow-up. Here, we use longitudinal binarized health attribute data from mice and humans that can be used to evaluate the FI to also quantify organismal damage and repair processes over time. As illustrated in the schematic in *Figure 1*, the change in number of deficits from one follow-up to the next is determined by the number of new deficits (indicating damage, with deficit values transitioning from 0 to 1, red arrow) minus the number of repaired deficits that were previously in a damaged state (with transitions of deficit values from 1 to 0, green arrow). These counts of damaged and repaired deficits between follow-ups represent summary measures of the underlying damage and repair processes. We model this process with a Bayesian Poisson model for counts of damaged and repaired deficits, using age-dependent damage and repair rates. For mice we use a joint longitudinal-survival model, which couples the damage and repair rates together with mortality. For humans, we use a similar model but without the survival component due to having no mortality data.

In our approach, damage rates are the probability of acquiring a new deficit per unit of time, and repair rates are the probability of repairing a deficit per unit time. These are aggregate measures of susceptibility to damage (lack of robustness), and ability to repair (resilience). The FI is a whole organism-level summary measure of health; accordingly, these aggregate damage and repair rates are also whole organism-level measures of robustness and resilience. Note that since these rates are per available deficit, repair rates may exceed damage rates while the FI is still increasing. This can occur due to relatively rapid repair per deficit of a small number of deficits, with a slower damage rate per deficit of a much larger number of undamaged attributes.

While damage of binarized health attributes with age necessarily follows from declining health, repair does not. However, almost all health attributes used in our mouse data have been previously shown to reverse either spontaneously or through extrinsic interventions such as drug treatments or lifestyle changes – see *Supplementary file 1*, with references. In *Figure 5—figure supplement 4* we

also show repair counts per deficit type. Nevertheless, for our data, not all deficits repair equally, and some rarely or never repair ('Cataracts', 'Tumours', 'diarrhea', and 'vaginal/uterine/penile prolapses').

## Both resilience and robustness decline in aging populations

We first establish the trends of repair and damage rates in aging. In *Figure 2*, we plot the age-dependence of the repair and damage processes in mice and humans for three mouse datasets (a) 1 *Keller et al., 2019*; (b) 2 *Bisset et al., 2022*; and (c) 3 *Schultz et al., 2020*; and (d) humans from the ELSA dataset (*Phelps et al., 2020*; *Steptoe et al., 2014*). Humans are plotted by decade of baseline age at entry to the study to separate out recruitment effects. Points are binned averages from the data, and lines are posterior samples from the model of the rates. Posterior predictive checks show good model quality, seen in *Figure 2—figure supplement 1* for mice (a-c) and humans (d).

In each of these datasets, there is a strong decrease in repair rates and increase in damage rates with age (except for damage rates in mouse dataset 2). Spearman rank correlations $\rho$ for each plot are also shown in *Figure 2*, highlighting the increase or decrease in rates with age, and 95% posterior credible intervals of these correlations are shown in brackets. Overall, we observe decreasing repair rates and increasing damage rates with age which signify decreasing resilience and robustness with age in both mice and humans. Decreasing repair and increasing damage both contribute to an increasing FI with age in mice and humans (shown in *Figure 2—figure supplement 2a–d*). We also observe higher FI scores in females versus males in both mice and humans, as reported previously (*Kane et al., 2019*; *Gordon and Hubbard, 2020*; *Kane and Howlett, 2021*).

We evaluate the contributions of damage and repair rates to survival using a joint longitudinal-survival model in mice. In *Figure 2e–g*, we show that damage rates have much larger hazard ratios for death than repair rates. These hazard ratios are for a fixed FI, itself a strong predictor of mortality in mice and in people (*Rockwood et al., 2017*), which shows that an increasing susceptibility to damage leads to larger decreases in survival than a comparable decline in resilience. Individuals survive longer when damage is avoided altogether, as compared to damage that is subsequently repaired. This intuitive result indicates that there may be lingering effects of the original damage and suggests that interventions that focus on robustness may be more effective than those that focus on resilience.

## The acceleration of damage accumulation is determined by a decline in robustness

The plots of FI vs. age shown in *Figure 2—figure supplement 2* (see also *Mitnitski et al., 2001*; *Mitnitski et al., 2005*; *Mitnitski et al., 2012*; *Mitnitski et al., 2013*) has a positive curvature, accelerating upwards near death (*Stolz et al., 2021*). This positive curvature is also seen in other summary measures such as Physiological Dysregulation (*Arbeev et al., 2019*). However, the origin of this curvature is unknown – whether it is due to a late-life decrease in resilience or a decline in robustness.

We measure the curvature of the FI with the second time-derivative, which can be computed with the age-slopes of the damage and repair rates (see Materials and methods). In *Figure 3*, we show the separate contributions to this curvature, separated into terms involving damage (pink) and terms involving repair (green). Summing these terms, we observe the typical positive curvature that indicates an acceleration of damage accumulation.

We find that the decline in robustness (indicated by the damage rate terms) has the strongest effect on the curvature of the FI. In mice, this is seen in every dataset and is significant at the indicated ages (*Figure 3b–d*) and for humans at older ages (*Figure 3e*). This observed effect indicates that it is the increase in damage with age, rather than the decline of repair, that causes this acceleration of net damage accumulation. The significance of this effect is evaluated by computing the posterior distribution of the difference between damage and repair terms. When at least 95% of this distribution is above/below zero, we take the effect to be significant. Credible intervals visualizing the difference are shown in *Figure 3—figure supplement 1*.

In *Figure 2e–g* we had shown that the decline in robustness has the strongest effect on survival in mice. Together with the results shown in *Figure 3*, these results highlight the important role of declining robustness in aging.

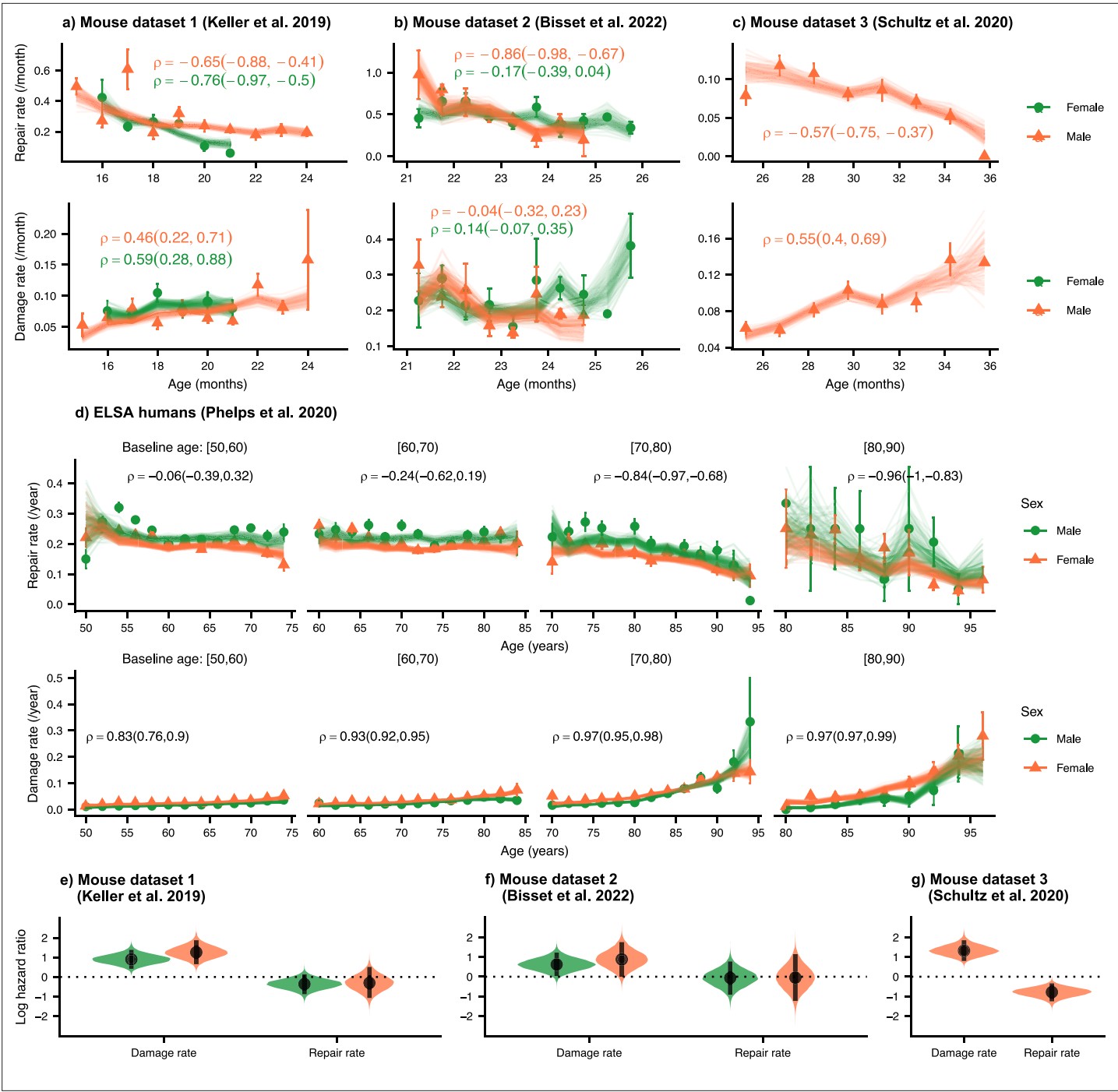

**Figure 2.** Repair rates decrease and damage rates increase with age. Repair rates vs age (top), damage rates vs age (bottom) for (**a**) Mouse dataset 1 (*Keller et al., 2019*), (**b**) Mouse dataset 2 (*Bisset et al., 2022*), (**c**) Mouse dataset 3 (*Schultz et al., 2020*) and (**d**) ELSA humans (*Phelps et al., 2020*; *Steptoe et al., 2014*) plotted by decades of baseline age. Points in all plots represent binned averages of rates from the data with standard errors, and lines represent posterior samples from Bayesian models of the rates (see Methods). For each plot, the mean Spearman's rank correlation $\rho$ between the rate and age is indicated by the median of the posterior and a 95% posterior credible interval in parenthesis. (**e**-**g**) Posterior distributions of log hazard ratios of death for damage and repair rates are shown as violin plots for the mouse datasets. These hazard ratios correspond to a 1 standard deviation increase in the damage or repair rates. The black interval shows a 95% credible interval around the median point.

The online version of this article includes the following source data and figure supplement(s) for figure 2:

**Source data 1.** Data used in figure.

**Figure supplement 1.** Posterior predictive check for joint models.

*Figure 2 continued on next page*

*Figure 2 continued*

**Figure supplement 1—source data 1.** Data used in figure.

**Figure supplement 2.** Increase in Frailty Index in mice and humans.

**Figure supplement 2—source data 1.** Data used in figure.

## Interventions modify damage and repair rates in mice and wealth correlates with rates in humans

Mouse datasets 1 (*Keller et al., 2019*) and 2 *Bisset et al., 2022* have additional intervention groups treated with either the ACE inhibitor enalapril, or voluntary aerobic exercise, respectively. In *Figure 4—figure supplement 1a and b*, we show that these interventions target both repair and damage processes, resulting in lower FI damage accumulation over time for the treated groups. In

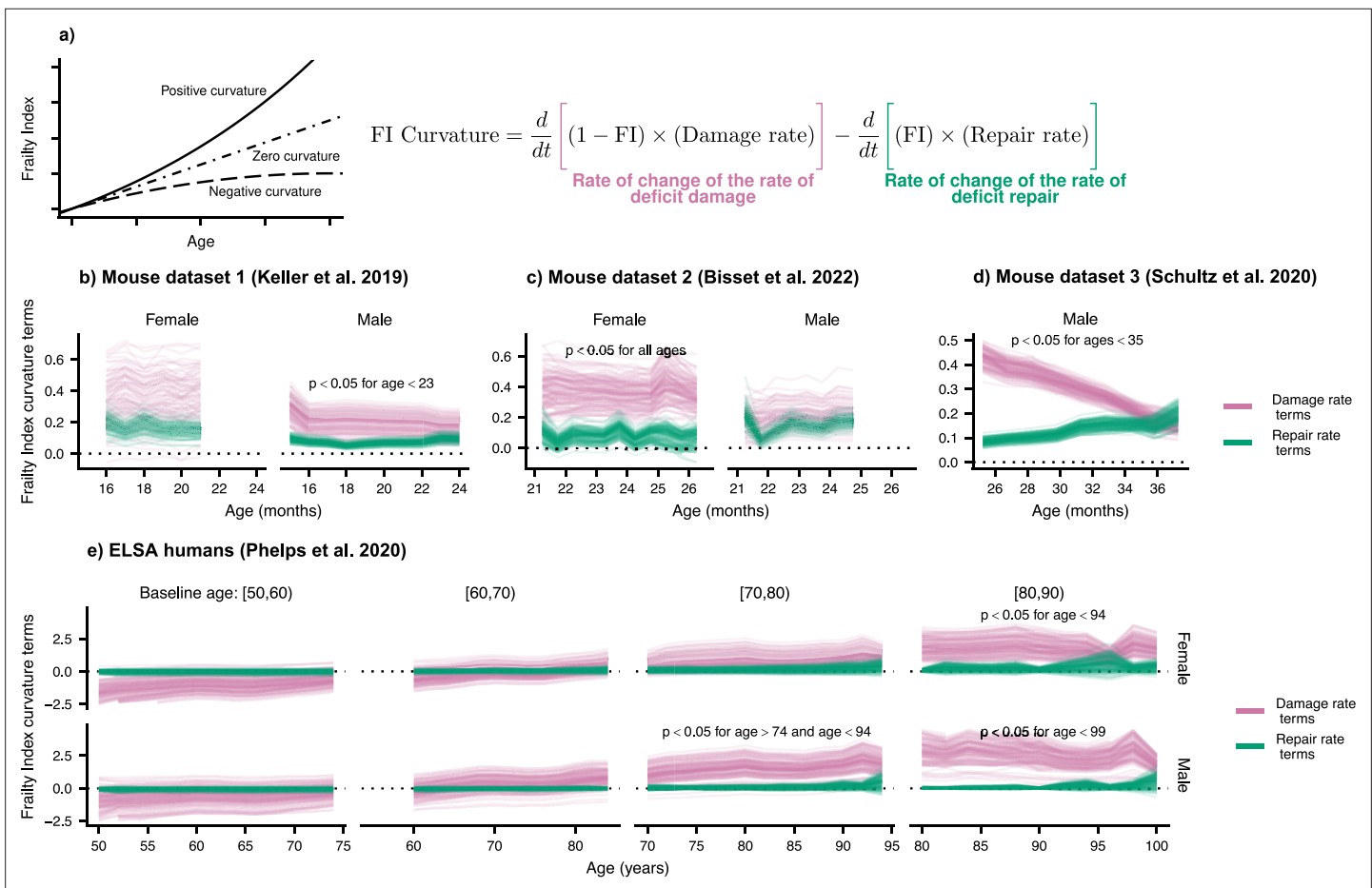

**Figure 3.** Frailty Index curvature is dominated by declining robustness. (**a**) Frailty Index curvature measures the rate of accumulation of damage. Positive curvature indicates an acceleration of damage accumulation, zero curvature indicates a constant accumulation of damage, and negative curvature indicates a decelerating accumulation of damage. Curvature is computed with the second time-derivative of the Frailty Index (Methods *Equation 22*). Terms of the curvature involving the repair rate (green) and the damage rate (pink) are separately shown. Lines represent posterior samples from our Bayesian models for (**b**) Mouse dataset 1 (*Keller et al., 2019*), (**c**) Mouse dataset 2 (*Bisset et al., 2022*), (**d**) Mouse dataset 3 (*Schultz et al., 2020*) and (**e**) ELSA humans (*Phelps et al., 2020*; *Steptoe et al., 2014*), plotted separately by decades of baseline age. On all plots, we indicate for which ages the proportion of the posterior for the difference in these terms that is negative is below 0.05; the Bayesian analogue of a p-value testing the contributions of robustness and resilience.

The online version of this article includes the following source data and figure supplement(s) for figure 3:

**Source data 1.** Data used in figure.

**Figure supplement 1.** Testing the effect of robustness, resilience, and interventions on curvature.

**Figure supplement 1—source data 1.** Data used in figure.

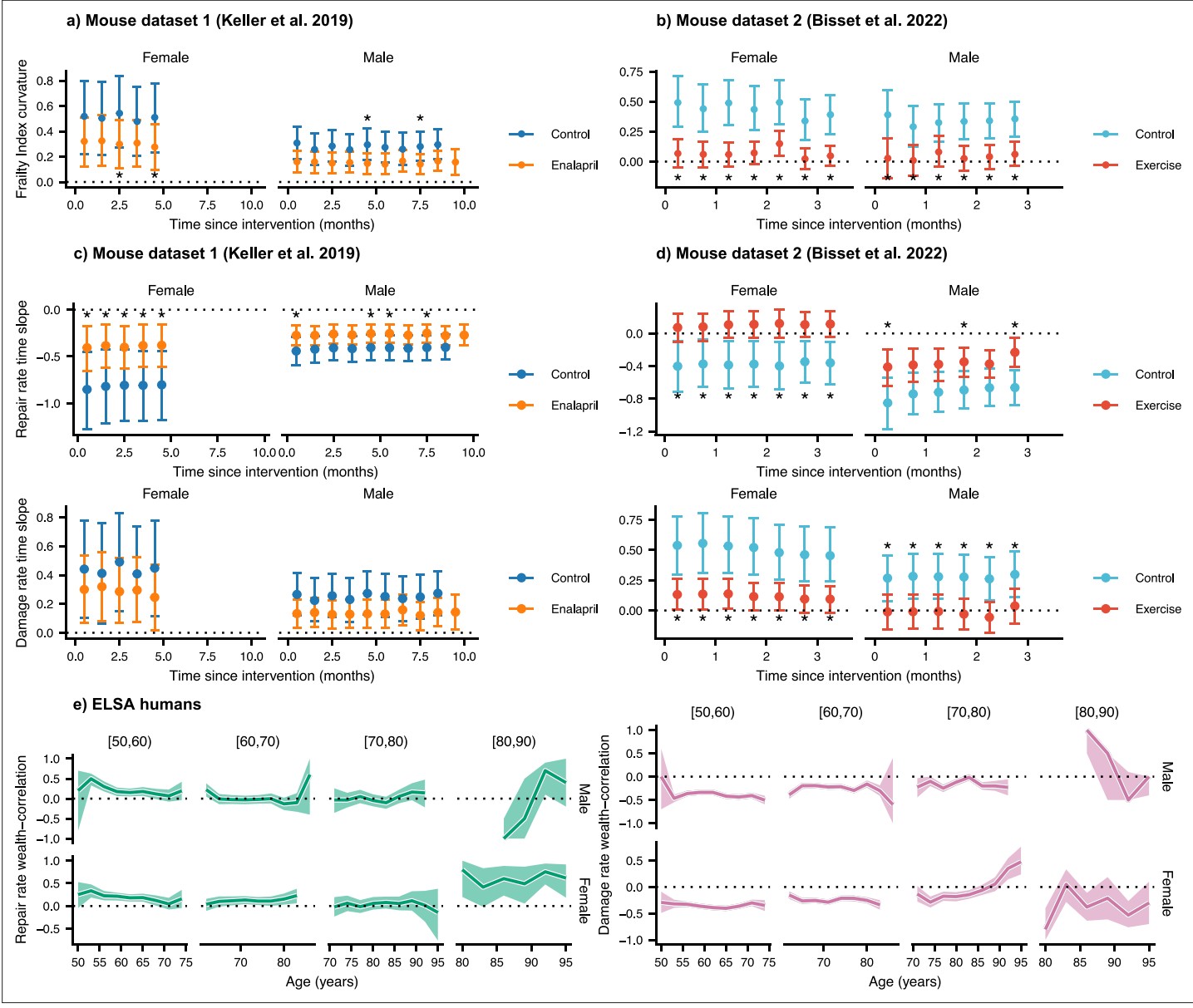

**Figure 4.** Interventions both increase resilience and decrease damage. (**a**, **b**) The effect of enalapril and exercise on Frailty Index curvature for mice for mouse datasets 1 and 2. 95% credible intervals for these curvatures are shown by errorbars around the median (point). Asterisks (*) indicate credible intervals for the difference between intervention and control fully exclude zero. (**c** ,**d**) Repair rates and damage rates time-slopes vs time since intervention for the effect of enalapril and exercise for mouse datasets 1 and 2. 95% credible intervals for these curvatures are shown by errorbars around the median (point). Asterisks (*) indicate credible intervals for the difference between intervention and control fully exclude zero. (**e**) Spearman rank correlation $\rho$ between wealth and repair rate (green) and damage rate (pink), vs age for ELSA humans. Individuals are separated by decades of baseline age, and 95% credible intervals for these correlations are shown as coloured regions around the median (thick line). We restrict this plot to ages with at least 3 individuals.

The online version of this article includes the following source data and figure supplement(s) for figure 4:

**Source data 1.** Data used in figure.

**Figure supplement 1.** The effect of interventions on repair, damage, and Frailty Index in mice.

**Figure supplement 1—source data 1.** Data used in figure.

**Figure supplement 2.** Humans stratified by terciles of household wealth.

**Figure supplement 2—source data 1.** Data used in figure.

*Figure 4a and b*, we investigate the effects of these interventions on the curvature of the FI. This curvature is strongly reduced by exercise in mouse dataset 2, with a weaker effect for enalapril the credible intervals of the intervention effects are shown in *Figure 3—figure supplement 1d and e*. Notably, exercise stops the acceleration in damage accumulation in both male and female mice by reducing the curvature to zero.

The effect of these interventions on the repair and damage rates is seen in *Figure 4c and d*, where 95% credible intervals for the age-slopes show the rate of increase or decrease of the repair and damage rates as age increases. These slopes include both the change in the rate with age, and the effect due to increasing FI with age. Interventions affect the rate of decrease of both repair and damage rates with time, resulting in less cumulative damage.

As shown in *Figure 4c*, enalapril attenuates the rate of decrease of repair rates in both male and female mice, resulting in age-slopes closer to zero than for controls. Significance is evaluated by computing the posterior distribution of the difference between control and intervention. Significance is shown with asterisks (*) when at least 95% of the distribution is above/below zero. In *Figure 4— figure supplement 1* we show a significant reduction in damage rate (but not slope) for male and female mice with enalapril. A sex-specific effect is seen for voluntary exercise. For female mice, voluntary exercise leads to stoppage of the decline in repair rates (to an approximately zero slope), whereas for male mice it just attenuates the decline (*Figure 4d*). For damage rates, female mice exhibit an attenuation of the rise with age whereas in male mice exercise stops the age-dependent rise exhibited by control mice.

For humans, we use net household wealth as a socioeconomic environmental factor that serves as a proxy for medical and behavioural interventions that are not individually tracked. This factor is not an intervention as in mice, and is simply correlational. As such we report correlations of wealth with repair and damage rates with age, rather than age-slopes after a specific intervention is initiated. In *Figure 4—figure supplement 2*, we show rates stratified by terciles of net household wealth, where the lowest tercile exhibits lower repair rates and higher damage rates for younger ages. Correspondingly, the FI is lower for individuals with a higher net household wealth. Treating the wealth variable as continuous, *Figure 4e* shows that repair rates are positively correlated with net household wealth, while damage rates are negatively correlated – with significant and stronger effects at younger ages. These results reinforce the findings in mice, where interventions impact both damage and repair rates. In humans, we also see some evidence of decreasing effects of wealth with age – although these may be confounded by recruitment effects depending on baseline age.

## Damage and repair have broad timescales

In the results above, we considered the average damage and repair transition rates vs age. Since individual deficits undergo stochastic transitions between damaged and repaired states, we can also measure the lifetime of these individual deficit states (see *Figure 5a*). These lifetimes are interval censored (transitions typically occur between observation times) and can be right-censored (death or drop-out before transition occurs). We use an interval censored-analogue to the standard Kaplan-Meier estimator for right censored data (see Maerials and methods) to estimate state-survival curves of individual damaged or repaired states. These state-survival curves in *Figure 5*, considering all possible deficits, represent the probability of a deficit remaining undamaged vs time since a repair transition, or remaining damaged vs time since a damage transition.

We generally observe a significant drop of state-survival probability at early times, indicating some rapid state transitions at or below the interval between measurements. However, all the curves also extend to very long times – towards the scale of organismal lifetime – indicating that both robustness and resilience operate over a broad range of timescales. These results highlight that repair can occur a long time after damage originally occurred. Note that the timescale of robustness as measured here is not robustness after a specific extrinsic stressor, but robustness from the implicit stressors of aging. A similar form of non-specific robustness has been measured in a previous study, using the onset age of disease (*Arbeev et al., 2019*).

As shown by exponential time-scales of resilience and robustness for individual deficits in *Figure 5— figure supplement 1* and *Figure 5—figure supplement 2*, mice deficits and human deficits exhibit a variety of time-scales of resilience and robustness. Some deficits repair soon after damage (or damage

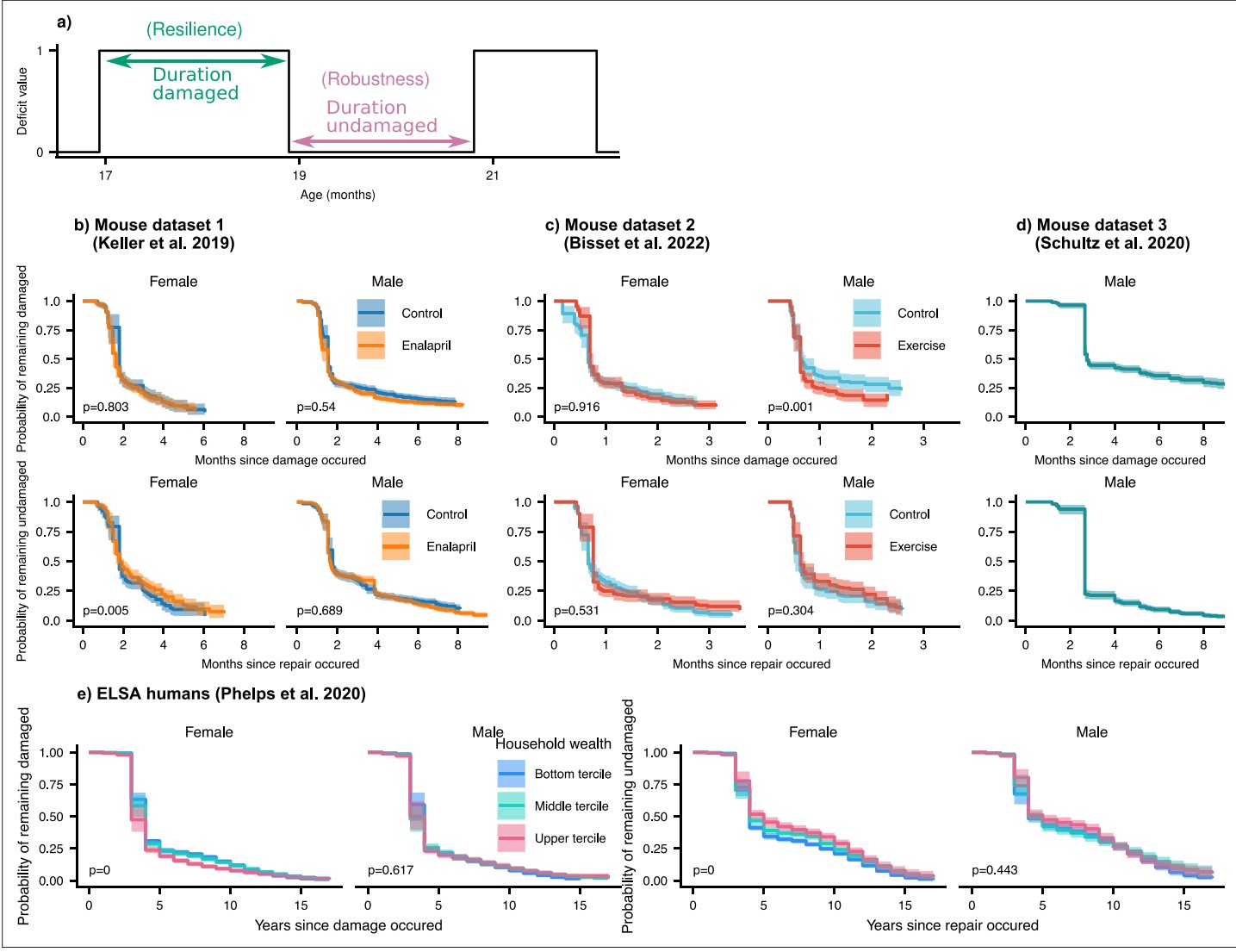

**Figure 5.** Resilience and robustness occur over both short and long time-scales in both mice and humans. (**a**) The time-scale of resilience is measured as the lifetime of the damaged state. The time-scale of robustness is measured as the lifetime of the undamaged state. Deficit state-survival curves, showing the probability of remaining in the current damaged or undamaged state since time of transition, are shown for (**b**) Mouse dataset 1, (**c**) Mouse dataset 2, (**d**) Mouse dataset 3, and (**e**) ELSA human dataset. The shaded regions are 95% posterior credible intervals. p-Values are shown on the lower left of each plot for generalized log-rank tests for the equality of the survival functions between the intervention or wealth groups (*Zhao et al., 2008* and *Zhaeo, 2012*).

The online version of this article includes the following source data and figure supplement(s) for figure 5:

**Source data 1.** Data used in figure.

**Figure supplement 1.** Time-scales of resilience for individual deficits.

**Figure supplement 2.** Time-scales of robustness for individual deficits.

**Figure supplement 2—source data 1.** Data used in figure.

**Figure supplement 3.** Sensitivity analysis of damage/repair.

**Figure supplement 3—source data 1.** Data used in figure.

**Figure supplement 4.** Sensitivity analysis of damage/repair.

**Figure supplement 4—source data 1.** Data used in figure.

soon after repair), and some repair (or damage) over a broad range of time-scales. The combination of all of these deficits result in the shape of the state-survival curves in *Figure 5*.

We evaluate the significance of the difference between state-survival curves with a generalized log-rank test (*Zhao et al., 2008*; *Zhaeo, 2012*). For the interventions studied, there are no dramatic changes of resilience or robustness timescales exhibited in mice. Exercise in the male mice slightly shifted the timescale of resilience, such that deficits were repaired faster in mice that were exercised compared to controls. We expect that we would observed stronger effects of the interventions on these time-scales if we had sufficient data to resolve the impact of the time at which the initial damage or repair event occurred – here we have grouped all times together. For humans (see *Figure 5e*), we see strong and significant effects on resilience and robustness timescales from household wealth in females, but not males. These effects are particularly strong for damage timescales, which characterize robustness: states remain healthy longer at higher wealth terciles.

## Discussion

We have presented a new approach for the assessment of damage (robustness) and repair (resilience) rates in longitudinal aging studies with binarized health attributes. With this approach, we have shown that both humans and mice exhibit increasing damage and decreasing repair rates with age, corresponding to decreasing robustness and resilience, respectively. We also demonstrate that decreasing robustness and resilience with age contribute to the acceleration of deficit accumulation for organisms. Decreasing robustness has approximately twice as large an effect when compared to declining resilience; decreasing robustness also has a stronger and significant effect on survival. While much of the focus in previous work has been on the decline of resilience in aging (*Ukraintseva et al., 2021*), our results indicate that decreasing robustness and decreasing resilience are both important processes underlying the increasing accumulation of health-related deficits with age, and the increasing rate of accumulation at older ages.

In the current study, the observed damage is assumed to occur due to natural processes, rather than a specific applied stressor (*Kirkland et al., 2016*; *Colón-Emeric et al., 2020*). Resilience measured by the observed repair also occurs without targeted interventions (certainly in mice, due to their absence of health-care), and so is likely to represent intrinsic resilience with respect to spontaneous damage due to the natural stressors of aging. Our approach has some similarities with recent approaches to measuring resilience by the autocorrelation timescale of intrinsic variations of continuous physiological state variables (*Pyrkov et al., 2021*; *Gijzel et al., 2017*; *Rector et al., 2021*). An advantage of our approach, which uses binarized variables, is that we can estimate both resilience and robustness using similar methods on the same data – so we can compare their relative effects. Previous work has modeled the change in the total count of discrete deficits with age (*Mitnitski et al., 2006*; *Mitnitski et al., 2007*; *Mitnitski et al., 2010*; *Mitnitski et al., 2012*; *Mitnitski et al., 2014*), but did not separately measure damage and repair. With our approach, we observe decreasing resilience and robustness with age in both mice and humans.

There are caveats to our approach. We may miss fast damage and repair dynamics that occur on time-scales shorter than the separation between observed time-points, for example, we cannot observe daily or weekly changes in deficit states in mice or monthly changes in humans. Therefore, our measurements of damage and repair can only be interpreted as the net damage and net repair between observed time-points. Furthermore, since we have defined damage and repair (and robustness and resilience) as average rates with respect to binarized attributes it remains an open question how they relate to damage and repair rates assessed from continuous health attributes. Our approach results in summary measures of damage and repair rates.

We are not aware of any selection bias in our mouse studies, and we applied joint modelling to mitigate survivor bias effects. To mitigate selection bias in the human data, we treated onset ages distinctly. There are nevertheless well-known 'healthy volunteer' effects that would bias the original population that we did not consider. Furthermore, we did not have human mortality data so we could not treat human survivor bias effects. Measurement errors could also contribute to both damage and repair rates – although presumably not in an age-dependent fashion. In contrast, we observe decreasing repair rates and increasing damage rates with age. Errors in deficit assessment are known to be small for mice (*Feridooni et al., 2017*; *Kane et al., 2017*). Supporting this, a sensitivity analysis

of pruning putative mouse measurement errors, shown in *Figure 5—figure supplement 3*, finds no qualitative changes.

We find that both damage and repair processes are targeted by interventions in mice. As a result, developing interventions to target either damage or repair separately is conceivable. While targeting either would affect net deficit accumulation, we found that the damage rate has a stronger effect on both mortality and the acceleration of damage accumulation than the repair rate. Consistent with this, recent work has shown that FI damage is also more associated with mortality than FI repair in humans (*Shi et al., 2021*). We predict that interventions that facilitate robustness (resistance to damage) may be more important at older ages, where damage accumulation normally accelerates. More broadly, rather than just targeting deficit accumulation or FI (*Howlett et al., 2021*), our results indicate that interventions could be improved by targeting an appropriate balance of damage and repair processes – in an age- and sex-dependent manner. Since both damage and repair occur on long timescales, this raises the possibility that these rates could be manipulated by interventions over a similarly broad range of timescales from the shortest times to organismal lifetimes. How to optimally deploy available interventions is not yet clear.

The effects of age on both damage and repair, in mice and humans, are qualitatively similar in male and female populations. Nevertheless, we have found that systemic interventions can have qualitatively distinct sex effects in mice. The ACE inhibitor enalapril has stronger effects in female mice. Voluntary exercise stopped the decline in repair rate with age for female mice, but not male mice, and stopped the increase in damage rate with age for male mice, but not female mice. These differences suggest that assessing both damage and repair rates, together with accumulated damage as a FI, in interventional aging studies can provide a clearer assessment of sex differences. Further studies are needed to tease out the sex-dependent effects of other aging interventions, and to provide quantitative insight into the mortality-morbidity paradox, where females live longer but have higher FI scores than males (*Kane and Howlett, 2021*; *Oksuzyan et al., 2008*).

Summary measures of health such as the FI exhibit an accelerating accumulation of health deficits with age (*Mitnitski et al., 2001*; *Mitnitski et al., 2005*; *Mitnitski et al., 2012*; *Mitnitski et al., 2013*). This universally observed behavior must be reflected in either increasing damage rates with age, decreasing repair rates, or – as we find – both. However, the question of whether, and by what mechanisms, damage and repair processes are coupled during aging remains unanswered. Both damage and repair rates have been typically modelled as functions of the health state in descriptive models of aging (*Taneja et al., 2016*; *Farrell et al., 2016*; *Farrell et al., 2018*), but without a mechanistic relationship between them apart from that imposed statistically by the observed accumulated damage. The precise relationships – and whether they are a universal feature of all aging organisms – remains to be determined. Studies of interventions should prove useful in this regard, because they can separately target damage and repair.

Our observations that repair timescales are broadly distributed, up to lifespan-scales, raise three fundamental questions for resilience studies. First, are interventions that facilitate recovery similarly effective after a broad range of timescales? This would imply that we may be able to target resilience with interventions over a longer timeframe than just acutely when damage occurs. Damage propagation may nevertheless limit the benefits of such late repair. Second, what determines the recovery timescales? As we have shown (*Figure 5—figure supplement 1*), different health attributes can have quite different recovery times. Third, would a similar broad range of resilience timescales be observed for challenge experiments with an induced stressor, and how might that depend on the magnitude and scale of the damage?

We have defined damage in terms of discrete transitions of dichotomized variables. It is possible that dichotomized deficits probe qualitatively different timescales than the continuous measures that are often considered in resilience studies. Future experimental resilience studies across a range of health attributes should explore longer timescales. It will also be important to assess how the broad range of recovery timescales we have uncovered compare to timescales extracted from autocorrelations of physiological state variables – which have also been limited to shorter times (*Pyrkov et al., 2021*).

We have also limited our study to 'clinical' variables at organismal scales. Further studies of resilience and robustness at different biological scales from cellular to organismal, with both continuous and discrete variables, and over organismal timescales, will help us to better understand how damage

and repair at cellular scales influences and is influenced by similar processes at organismal scales. It is easier to conceive of how damage can propagate from cellular to organismal (*Howlett and Rockwood, 2013*), but harder to conceptualize how cellular repair processes such as DNA repair pathways and autophagy (*Kirkwood, 2011*) might similarly propagate.

Most of the mouse health deficits analyzed here have previously been shown to reverse either spontaneously or in response to interventions including drug treatment or exercise (*Supplementary file 1*). There are only a few deficits that rarely or never repaired in the current study: cataracts, tumours, diarrhea, and prolapses. Both this study and the literature suggest that most deficits that make up the mouse frailty index can be reversed or repaired. Investigation of the specific mechanisms responsible for the spontaneous repair of deficits, and how they scale from the cell to the organism, should be the focus of future work. We speculate that spontaneous repair occurs by the same pathways that are targeted by health interventions. It is likely that deficits reverse if interventions target one or more of the molecular/cellular pillars of aging, including macromolecular damage, dysregulated stress response, disruption in proteostasis, metabolic dysregulation, epigenetic drift, inflammaging, and stem cell exhaustion (*Goh et al., 2022*). In terms of the specific interventions investigated here, previous studies have shown that beneficial effects of enalapril treatment and exercise on frailty are attributable, at least in part, to effects on chronic inflammation (*Bisset et al., 2022*; *Keller et al., 2019*).

The increasing availability of longitudinal health data over the lifespan of model aging organisms facilitates the analysis of damage and repair rates, and how they extend and change over the organismal lifespan. These damage and repair rates underlie the accumulation of damage that describes aging. Here we have shown the value of considering both resilience and robustness over the lifespan. Further studies will be able to determine how widespread organismal and sex differences in these effects are, and how universal they may prove to be. Studies of the effects on damage and repair rates of both targeted and systemic interventions will also be crucial. We have studied only three interventions or conditions so far (e.g. enalapril and exercise in mice, and wealth in humans). There are many other possibilities, including treatment with geroprotectors (*Gonzalez-Freire et al., 2020*) and lifestyle interventions, that could be deployed both in humans and in aging animal models.

# Materials and methods
## Mouse data
For the mouse portion of this manuscript, published data on longitudinal health-related deficits in C57BL/6 mice from three papers was used (*Keller et al., 2019*; *Bisset et al., 2022*; *Schultz et al., 2020*). A brief summary of the methods of each paper is below.

### Mouse dataset 1 (*Keller et al., 2019*)
Male and female C57BL/6 mice were assessed for deficits approximately every 4 weeks from 16 to either 21 months of age (females) or 25 months of age (males). Mice were fed either a diet containing enalapril (280 mg/kg) or control diet for the duration of the experiment. After pre-processing (below), this data contains 21 female control mice, 25 female enalapril mice, 13 male control mice, and 25 male enalapril mice.

### Mouse dataset 2 (*Bisset et al., 2022*)
Male and female C57BL/6 mice were assessed for deficits approximately every 2 weeks from 21 to 25 months of age. Mice were all singly housed, and half were provided a running wheel for voluntary exercise. After pre-processing (below), this data contains 11 female control mice, 11 female exercise mice, 6 male control mice, and 6 male exercise mice.

### Mouse dataset 3 (*Schultz et al., 2020*)
Male C57BL/6Nia mice were assessed for deficits approximately every 6 weeks from 21 months of age until their natural deaths. After pre-processing (below), this data contains 44 male control mice.

### Mouse clinical frailty index assessment
Each of the papers above assessed health deficits using the mouse clinical frailty index as described previously (*Whitehead et al., 2014*). Briefly, this assessment involves scoring 31 non-invasive

health-related measures in mice. Most measures are scored as 1 for a severe deficit, 0.5 for a moderate deficit and a 0 for no deficit. Deficits in body weight and temperature were scored based on deviation from reference values in young adult animals, such that a difference of less than 1 SD was scored 0, a difference of ±1 SD was scored 0.25, a difference of ±2 SD was scored 0.5, a difference of ±3 SD was scored 0.75, and a difference of more than 3 SD received the maximal deficit value of 1 (**Whitehead et al., 2014**). The deficits of malocclusions and body temperature were not assessed in mouse group 3 (**Schultz et al., 2020**), leaving only 29 deficits for this dataset.

The variables in the Frailty Index are, 'Alopecia', 'Fur color loss', 'Dermatitis', 'Coat condition', 'Loss of whiskers', 'Kyphosis', 'Distended abdomen', 'Vestibular disturbance', 'Cataracts/corneal capacity', 'Eye discharge/swelling', 'Microphthalmia', 'Malocclusions', 'Rectal prolapse', 'Penile prolapse', 'Mouse grimace scale', 'Piloerection', 'Tail stiffening', 'Gait', 'Grip strength', 'Body condition', 'Hearing loss', 'Vision loss', 'Menace reflex', 'Tremor', 'Tumors/growths', 'Nasal discharge', 'Diarrhoea', 'Breathing rate/depth', 'Body temperature', 'Body weight'. The FI reference sheet at https://github.com/SinclairLab/frailty, shows examples of mice corresponding to the different levels of the deficits.

## Mouse data pre-processing

For mouse dataset 1, we impute missing deficit values by propagating the last observed value forward. If the first observed deficit is missing, it is imputed by propagating the first observed value backward. Less than 1% of all total deficit values are missing in this dataset. No values in the other datasets are missing.

Deficits are scored on a fractional scale, with deficit $i$ having values $d_i \in \{0, 0.25, 0.5, 0.75, 1\}$. To treat these as binary, we represent each fractional deficit $d_i$ by 4 ordered binary deficits, $[d_i^{(1)}, d_i^{(2)}, d_i^{(3)}, d_i^{(4)}]$. Fractional deficits are then represented by setting $4 \times d_i$ of these ordered binary deficits to 1. For example if $d_i = 0.75$, this is represented as $[1, 1, 1, 0]$.

A new Frailty Index is then created by taking all of these new binary deficits, representing a $4 \times 31 = 124$ item Frailty Index ($4 \times 29 = 116$ for mouse dataset 3). This process preserves the FI scores, and a single repair or damage transition on this scale can be interpreted as taking a step of size 0.25 on the fractional deficit scale. Each step of a deficit originally on the $[0, 0.5, 1]$ scale corresponds to 2 steps of size 0.25 on this new scale.

Measurement times with abnormally short or long intervals are removed. In mouse dataset 2, measurement times less than 0.1 months from the previous time are removed. In mouse dataset 3, measurement times more than 2 months from the previous time are removed.

In each dataset, mice with less than 2 observed time-points are removed.

## Human data and pre-processing

We use human data from the English Longitudinal Study of Aging (**Phelps et al., 2020**; **Steptoe et al., 2014**). We select individuals that have full data for net household wealth and activities of daily living (ADL) and instrumental activities of daily living (IADL). A Frailty Index is created from the count of 10 possible ADLs and 13 possible IADLs, giving a fraction out of 23. Each of these variables are binary with values $\{0, 1\}$.

The ADLs are 'Have difficulty': 'Walking 100 yards', 'Sitting for about two hours', 'Getting up from a chair after sitting for long periods', 'Climbing several flights of stairs without resting', 'Climbing one flight of stairs without resting', 'Stooping, kneeling, or crouching', 'Reaching or extending arms above shoulder level', 'Pulling/pushing large objects like a living room chair', 'Lifting/carrying over 10 lbs, like a heavy bag of groceries', and 'Picking up a 5 p coin from a table'. The IADLs are 'Have difficulty': 'Dressing, including putting on shoes and socks', 'Walking across a room', 'Bathing or showering', 'Eating, such as cutting up your food', 'Getting in or out of bed', 'Using the toilet, including getting up or down', 'Using a map to get around a strange place', 'Preparing a hot meal', 'Shopping for groceries', 'Making telephone calls', 'Taking medications', 'Doing work around the house or garden', and 'Managing money, e.g. paying bills and keeping track of expenses'.

We restrict individuals to those that were recruited to the study between the ages of 50 and 89 years. We drop individuals with follow-up time intervals above 4 years and individuals with fewer than 6 follow-ups. This removes 15399 individuals from the dataset, 4326 of which only had a single

time-point, 2291 had 2 time-points, 2095 had 3 time-points. The final selected individuals are followed for between 13 and 18 years, with 60% of the individuals being followed for 16 years.

We use net household wealth, as determined in the financial assessment in wave 5 of the ELSA data. We drop individuals that have parts of this assessment imputed. The raw value of net household wealth spans several orders of magnitude (and includes negatives for individuals in debt), and so is transformed by $w = \log\left(w_{\text{raw}} + \text{mean}(w_{\text{raw}})\right)$.

After pre-processing, this data contains 1049 males and 1300 females with time-intervals of approximately 2 years between observations. There are 1222 individuals from baseline ages in $[50, 60]$, 827 with baseline ages in $[60, 70]$, 281 with baseline ages in $[70, 80]$, and 19 with baseline ages in $[80, 90]$.

## Extracting damage and repair counts

In each dataset, we observe the state of $N$ binary health deficits $\{d_{jk}\}_{k=1}^{N}$ for each subject at a set of observation times $\{t_j\}_{j=1}^{J}$. Summing up the number of deficits at each time, we get deficit counts for each observation time, $\{n_j\}_{j=1}^{J}$, which is used to compute the Frailty Index $f_j = n_j/N$.

We compute the number of deficits damaged ($0 \to 1$ transitions) and repaired ($1 \to 0$ transitions) between two time points $t_j$ and $t_{j+1}$, denoted as $n^d(t_j)$ or $n^r(t_j)$. These counts satisfy $n(t_{j+1}) = n(t_j) + n^d(t_j) - n^r(t_j)$, linking these damage and repair processes with the Frailty Index.

## Modelling

We model deficit repair and damage as Poisson point processes with time-dependent rates, $\lambda^r(t)$ and $\lambda^d(t)$. The count of deficits repaired or damaged in an interval $[t_1, t_2]$ is assumed to follow a Poisson distribution, with mean equal to the instantaneous rate $\lambda^r(t)$ or $\lambda^d(t)$ integrated over this interval times the number of possible deficits available to be repaired $\Lambda^r(t) = \int \lambda^r(t)n_t dt$, or damaged $\Lambda^d(t) = \int \lambda^d(t)(N - n_t)dt$. For computational convenience, we use constant rates within each time-interval to approximate these integrals, $\Lambda^r(t) \approx \lambda^r(t)n_t \Delta t$ or $\Lambda^d(t) \approx \lambda^d(t)(N - n_t)\Delta t$.

We perform Bayesian inference on our models by inferring the posterior distribution of the parameters given the observed data.

### Joint longitudinal-survival model for mice data

We use a joint modelling framework to model repair and damage rates, while assessing their effect on survival. We decompose the multivariate joint distribution of the observed longitudinal damage and repair counts and survival outcome by coupling survival with the repair and damage rates $\lambda_i^r(t)$ and $\lambda_i^d(t)$ (**Hickey et al., 2016**; **Brilleman et al., 2020**),

$$p(T_i, c_i, \{n_i^r(t)\}, \{n_i^d(t)\} | \lambda_i^r(t), \lambda_i^d(t)) = p(T_i, c_i | \lambda_i^r(t), \lambda_i^d(t)) p(\{n_i^r(t)\}, \{n_i^d(t)\} | \lambda_i^r(t), \lambda_i^d(t)).$$

We indicate final follow-up times for individual $i$ as $T_i$ with a censoring indicator $c_i$ where 1 is censored and 0 is an observed death.

### Longitudinal component

We use a linear Poisson model for the longitudinal damage and repair rates. A Softplus function, $\log(1 + e^x)$, is used to enforce positive rates. This function is chosen because it is approximately linear for larger values of $x$, in contrast to $e^x$ which is often used for Poisson models (which resulted in poor behaviour for our models). The form of this model is,

$$\lambda_i^r(t) = \text{Softplus}\left(\beta^r \cdot \mathbf{x}_i(t) + b_{i,0}^r + b_{i,1}^r t\right), \tag{1}$$

$$\lambda_i^d(t) = \text{Softplus}\left(\beta^d \cdot \mathbf{x}_i(t) + b_{i,0}^d + b_{i,1}^d t\right), \tag{2}$$

$$n_i^r(t_j) | \lambda_i^r(t_j), n_i(t_j) \sim \text{Poisson}\left(n_i(t_j)\lambda_i^r(t_j)(t_{j+1} - t_j)\right), \tag{3}$$

$$n_i^d(t_j) | \lambda_i^d(t_j), n_i(t_j) \sim \text{Poisson}\left((N - n_i(t_j))\lambda_i^d(t_j)(t_{j+1} - t_j)\right), \tag{4}$$

$$n_i(t_{j+1}) = n_i(t_j) + n_i^d(t_j) - n_i^r(t_j). \tag{5}$$

The first two equations describe the time-dependent repair and damage rates, $\lambda^r(t)$ and $\lambda^d(t)$. These rates represent the probability of repair or damage, per deficit per unit time. These rates are multiplied

by the number of deficits that can repair $n(t_j)$ or the number that can damage $N - n(t_j)$ and the time-interval $t_{t_{j+1}} - t_j$ to compute the mean count of repaired or damaged deficits for Poisson distributions. The last *Equation 5* shows how we can compute the total count of deficits from this model, allowing this model to be used to model the Frailty Index as well.

The full-cohort parameters are denoted $\beta$ and the subject-specific intercept and time-slopes $b_{i,0}, b_{i,1}$. The variables $\mathbf{x}_i(t)$ include the covariates and their interactions with sex and intervention group,

$$\mathbf{x}_i(t) = (1, t, \text{sex}, \text{treatment}, f, a_0, \text{sex} \times \text{treatment}, \text{sex} \times t, \text{treatment} \times t, \text{sex} \times \text{treatment} \times t).$$

The "treatment" variable is a 0/1 indicator for enalapril in mouse group 1 or exercise in mouse group 2. The other variables are the time from baseline $t$, the Frailty Index $f$, baseline age $a_0$, and sex (M/F). These interactions allow sex and intervention group specific time-slopes.

The repair and damage processes are linked by including correlations between the subject-specific parameters $[\boldsymbol{b}_i^r, \boldsymbol{b}_i^d] \sim \mathcal{N}(0, \boldsymbol{\Sigma})$.

## Survival component

We jointly model these repair and damage processes with survival, with proportional hazards and a baseline hazard parameterized with M-splines (*Ramsay, 1988*) (which are always non-negative). The damage and repair processes are linked with survival by including damage rate and repair rate in the hazard rate,

$$h_i(t) = h_0(t, \text{sex}) \exp\left(\gamma \cdot \mathbf{u}_i(t) + \gamma^r \text{Softplus}^{-1} \lambda_i^r(t) + \gamma^d \text{Softplus}^{-1} \lambda_i^d(t)\right), \tag{6}$$

$$h_0(t) = (\text{male}) \cdot \sum_{l=1}^{L} a_{l,\text{male}} M_{l,3}(t|\mathbf{k}) + (\text{female}) \cdot \sum_{l=1}^{L} a_{l,\text{female}} M_{l,3}(t|\mathbf{k}), \quad \sum_{l=1}^{L} a_l = 1, \quad a_l \geq 0,$$

$$S_i(t) = \exp\left(-\int_{t_0}^{t} h_i(s) ds\right). \tag{7}$$

The first equation describes the hazard rate $h_i(t)$ in terms of the covariates $\mathbf{u}_i$ and the repair and damage rates. The baseline hazard $h_0(t, \text{sex})$ is modeled with sex-specific splines, due to the large disparity in survival by sex. The covariates are $\mathbf{u}_i = (1, \text{sex}, \text{treatment}, \text{sex} \times \text{treatment}, f, a_0)$. The $\{M_{l,3}(t|\mathbf{k})\}_{l=1}^{L}$ functions are M-spline basis functions of order 3 with an $L$-component knot vector $\mathbf{k}$.

## Priors and hyperparameters

We use weakly informative priors to regularize parameters,

$$\beta_0^r, \ \beta_0^d, \ \gamma_0 \sim \mathcal{N}(0, 3), \quad \boldsymbol{\beta}^r, \ \boldsymbol{\beta}^d, \ \boldsymbol{\gamma}, \ \gamma^r, \ \gamma^d \sim \mathcal{N}(0, 1), \tag{8}$$

$$[\mathbf{b}_i^r, \ \mathbf{b}_i^d] \sim \mathcal{N}(0, \boldsymbol{\Sigma}), \quad \boldsymbol{\Sigma} = \boldsymbol{\sigma}\boldsymbol{\Omega}\boldsymbol{\sigma}, \quad \boldsymbol{\sigma} \sim \text{HalfCauchy}(0, 1), \quad \boldsymbol{\Omega} \sim \text{LKJ}(2), \tag{9}$$

$$\mathbf{a} \sim \text{Dirichlet}(1.0, L = 17), \quad \mathbf{k} = \{\min(\{T_i\}_i), Q_{0.05}(\{T_i\}_i), ..., Q_{0.95}(\{T_i\}_i), \max(\{T_i\}_i)\}. \tag{10}$$

Broad $\mathcal{N}(0, 3)$ priors are used on intercept parameters and narrow $\mathcal{N}(0, 1)$ priors on covariate coefficients (*Equation 8*). The covariance matrix $\Sigma$ for the coupling of the subject-specific parameters $\mathbf{b}_i$ is decomposed in terms of a correlation matrix $\Omega$ with an LKJ prior and standard deviations $\sigma$ with half-Cauchy distributions (*Equation 9*). The LKJ distribution is a standard prior for correlation matrices, where $\text{LKJ}(\eta = 1)$ is a uniform distribution over correlation matrices (*Lewandowski et al., 2009*). Increasing $\eta$ results in a sharper peak at the identity matrix.

These weak priors have the effect of making large parameter values unlikely (all parameters are put on the same scale by standardizing the values of all covariates first to mean 0 and variance 1), which improves the computational speed of the MCMC sampler. Choice of such weak priors affect quantitative results to a small extent, but do not affect our qualitative results – like other hyperparameters.

Spline coefficients $\mathbf{a}$ use a Dirichlet distribution with concentration 1, representing a uniform prior on the simplex $\sum_{l=1}^{L} a_l = 1$, $a_l \geq 0$. We use $L = 17$ spline knots with knots at the minimum last follow-up age, the maximum, and 15 uniformly spaced quantiles from 0.05 to 0.95 of the last follow-up age (*Equation 10*).

Integrals of the hazard rate are computed with 5-point Gaussian Quadrature between each observed time interval.

## Non-linear modeling for human data

There is much more human data than mice and the data is more complex, where linear effects are not sufficient to capture the combined influence of wealth, baseline age, and time. We use a non-linear Poisson model with non-constant coefficients to include additional degrees of freedom. We parameterize these non-constant coefficients with B-splines. The individuals selected from ELSA with wealth data do not have mortality data available, simplifying the model from the joint model used above for mice.

Our model has the form,

$$\lambda_i^r(t) = \text{Softplus}\left(\beta_0^r \cdot \mathbf{x}_i(t) + \beta_1^r(w, a_0) + \beta_2^r(w, a_0) \times \text{sex} + \beta_3^r(w, a_0) \times t + \beta_4^r(w, a_0) \times \text{sex} \times t + b_{i,0}^r\right), \quad (11)$$

$$\lambda_i^d(t) = \text{Softplus}\left(\beta_0^d \cdot \mathbf{x}_i(t) + \beta_1^d(w, a_0) + \beta_2^d(w, a_0) \times \text{sex} + \beta_3^d(w, a_0) \times t + \beta_4^d(w, a_0) \times \text{sex} \times t + b_{i,0}^d\right), \quad (12)$$

$$n_i^r(t_j)|\lambda_i^r(t_j), n_i(t_j) \sim \text{Poisson}\left(n_i(t_j)\lambda_i^r(t_j)(t_{j+1} - t_j)\right), \quad (13)$$

$$n_i^d(t_j)|\lambda_i^d(t_j), n_i(t_j) \sim \text{Poisson}\left(N - n_i(t_j)\lambda_i^d(t_j)(t_{j+1} - t_j)\right), \quad (14)$$

where $w$ denotes wealth and $a_0$ denotes baseline age and,

$$\mathbf{x}_i(t) = (1, t, \text{sex}, w, f, a_0, \text{sex} \times t, w \times t, a_0 \times t, \text{sex} \times f, w \times f, a_0 \times f). \quad (15)$$

The non-constant coefficients $\{\beta_k(w, a_0)\}_k$ are implemented as 2D B-splines for wealth and baseline age with 5 wealth knots and 5 baseline age knots at the minimum, maximum and terciles of these variables. We use smoothing 2D random-walk priors on the spline coefficients,

$$s_{11}, \tau_w, \tau_{b_0} \sim \mathcal{N}(0, 1), \quad p_w, p_{b_0} \sim \text{Dirichlet}(1.5), \quad (16)$$

$$s_{ij} \sim p_w \mathcal{N}(s_{i-1,j}, \tau_w) + p_{b_0} \mathcal{N}(s_{i,j-1}, \tau_{b_0}), \quad (17)$$

$$\beta_k(w, a_0) = \sum_{i,j=1}^{5} s_{ij} B_{i,3}(w; \mathbf{k}_w) B_{j,3}(a_0; \mathbf{k}_{a_0}). \quad (18)$$

All other priors are the same as in the mouse modelling.

Note, in the human data we do not include subject-specific time-slopes $b_{i,1}^r$ and $b_{i,1}^d$ as we did in the mouse data, since we have much shorter time-series. When these slopes are included, we see evidence of the model over-fitting to the data by the proportion of residuals including zero within 95% credible intervals being much higher than 0.95 – nearing 0.99 to 1.00.

## Derivatives

We can compute the derivative of the Frailty Index according to the modelled repair and damage rates,

$$\frac{d}{dt} f_i(t) = (1 - f_i)\lambda_i^d(t) - f_i\lambda_i^r(t). \quad (19)$$

To understand the effect of interventions, we compute the derivative with respect to time for the repair and damage rates,

$$\frac{d}{dt}\lambda_i^r(t) = \frac{\partial \lambda_i^r(t)}{\partial t} + \frac{\partial \lambda_i^r(t)}{\partial f}\frac{df_i(t)}{dt},$$
$$= \left(\beta^r \cdot \frac{d\mathbf{x}_i(t)}{dt} + \beta^r \cdot \frac{d\mathbf{x}_i(t)}{df} + b_{i,1}^r\right)\frac{e^{\lambda_i^r(t)}}{e^{\lambda_i^r(t)}+1}. \quad (20)$$

$$\frac{d}{dt}\lambda_i^d(t) = \frac{\partial \lambda_i^r(t)}{\partial t} + \frac{\partial \lambda_i^r(t)}{\partial f}\frac{df_i(t)}{dt},$$
$$= \left(\beta^d \cdot \frac{d\mathbf{x}_i(t)}{dt} + \beta^d \cdot \frac{d\mathbf{x}_i(t)}{df} + b_{i,1}^D\right)\frac{e^{\lambda_i^d(t)}}{e^{\lambda_i^d(t)}+1}. \quad (21)$$

This is the slope of these rates vs time, with the increase in the Frailty Index $f(t)$ included. While we only include explicit linear effects of time in the model, the increase in Frailty Index with time can influence the derivative to change.

We can compute the curvature as the second derivative of the Frailty Index with age, written in terms of first derivatives of the rates,

$$\frac{d^2}{dt^2} f_i(t) = \left[(1 - f_i(t))\frac{d\lambda_i^d(t)}{dt} - \frac{df_i(t)}{dt}\lambda_i^d(t)\right] - \left[f_i(t)\frac{d\lambda_i^r(t)}{dt} + \frac{df_i(t)}{dt}\lambda_i^r(t)\right]. \quad (22)$$

The first group of terms are those involving damage rate (robustness) and the second group of terms are those involving repair (resilience). These terms are plotted in *Figure 3*.

## Repair and damage timescale mice and human data

We observe the amount of time that has passed between damage and repair events, and vice versa. This can be used to determine the time-scales of these damage and repair processes. However, since a deficit might damage and the individual dies before the deficit is ever repaired, there is right censoring. Additionally, since observations are only made at specific time-points so that we cannot determine the exact time at which a deficit damaged or repaired, there is interval censoring. To estimate the distribution of repair and damage times, we treat repair and damage events for each deficit as a mixture of interval and right censored events (*Zhao et al., 2008*; *Zhaeo, 2012*). Accordingly, we model state-survival curves for the damaged state (time-scale of resilience) and undamaged state (time-scale of robustness).

We use a Bayesian survival model with M-splines for the baseline hazard,

$$h(t) = e^{\gamma_0} \sum_{l=1}^{L} a_l M_{l,3}(t|\mathbf{k}), \qquad \sum_{l=1}^{L} a_l = 1, \qquad S(t) = \exp\left(-\int_{t_0}^{t} h(s)ds\right). \tag{23}$$

This is fit separately for sex and control/intervention groups.

We include both interval-censoring and right-censoring in the likelihood for individual $i$,

$$p(T_i^{\text{Lower}}, T_i^{\text{Upper}}, T_i, c_i|\{a_l\}_l, \gamma_0) = [S(T_i)]^{c_i}[S(T_i^{\text{Lower}}) - S(T_i^{\text{Upper}})]^{1-c_i}, \tag{24}$$

where $T^{\text{Lower}}$ is the lower interval bound, $T^{\text{Upper}}$ is the upper interval bound, $T$ is a time of right censoring, and $c$ is the right censoring indicator (c=1 event censored, c=0 event occurs). We use 32 knots set at 30 evenly spaced quantiles of the event times from 0.1 to 0.9 together with the minimum and maximum event time. A uniform $\text{Dirichlet}(1.0, 32)$ prior is used for the spline coefficients and a broad $\mathcal{N}(0, 10)$ prior for $\gamma_0$.

## MCMC sampling

We use the STAN no U-turn sampler (NUTS) (*Stan Development Team, 2020*). We use 4000 warm-up iterations and 6000 sampling iterations on four separate chains for the mouse joint models. For mouse dataset 2, we use the sampler settings adapt_delta = 0.95, max_treedepth = 20 to avoid divergent sampler transitions. For the human model, we use two separate chains with 1000 warm-up iterations and 3000 sampling iterations. For the interval-censored Bayesian survival models, we use 2000 warm-up iterations and 3000 sampling iterations for four separate chains. Number of sampling iterations are chosen to achieve adequate effective sample sizes, while remaining computationally feasible.

In *Figure 2—figure supplement 1* we perform posterior predictive checks (*Gabry et al., 2019*; *Gelman et al., 2020*) for the mice and human models by plotting observed and simulated distributions of counts. We also compute $R^2$ statistics (*Vehtari et al., 2021*) and the coverage of credible intervals for residuals.

## Sensitivity analysis

Some of the observed damage and repair transitions may be due to measurement error or data entry errors. In particular, we believe this may be the case for some of the isolated damage and repair transitions. For example, if we consider 5 time-points where a variable has values $\{0, 0, 1, 0, 0\}$, this may be an erroneous transition.

Under the assumption that such erroneous transitions will most likely occur isolated from true damage/repair, we prune the data by removing these transitions, e.g. $\{0, 0, 1, 0, 0\}$ becomes $\{0, 0, 0, 0, 0\}$ (erroneous damage) or $\{1, 1, 0, 1, 1\}$ becomes $\{1, 1, 1, 1, 1\}$ (erroneous repair). We only prune damage/repair events isolated by 2 of the opposite state on either side, as shown here. In *Figure 5—figure supplement 3*, we show that pruning these values only has a limited effect.

## Code and data availability

Our code for data pre-processing, modelling, and plotting is available https://github.com/Spencerfar/aging-damagerepair, (*Farrell, 2022* copy archived at swh:1:rev:4fe6f883d37dda6b-9059c53aa9366f4ff2665a43). Human ELSA data can be accessed by agreeing to an End User Licence https://www.elsa-project.ac.uk/accessing-elsa-data and downloading waves 1–9. Mouse dataset 3 is freely available at https://github.com/SinclairLab/frailty, (*Sinclair Lab, 2022*; *Schultz et al., 2020*).

## Additional information

### Funding

| Funder | Grant reference number | Author |
|---|---|---|
| Natural Sciences and Engineering Research Council of Canada | RGPIN-2019-05888 | Andrew D Rutenberg |
| Canadian Institutes of Health Research | PJT 155961 | Susan E Howlett |
| Heart and Stroke Foundation of Canada | G-22-0031992 | Susan E Howlett |

The funders had no role in study design, data collection and interpretation, or the decision to submit the work for publication.

### Author contributions

Spencer Farrell, Conceptualization, Data curation, Software, Formal analysis, Investigation, Visualization, Methodology, Writing – original draft, Writing – review and editing; Alice E Kane, Conceptualization, Data curation, Writing – review and editing; Elise Bisset, Data curation, Writing – review and editing; Susan E Howlett, Conceptualization, Supervision, Funding acquisition, Project administration, Writing – review and editing; Andrew D Rutenberg, Conceptualization, Supervision, Funding acquisition, Methodology, Project administration, Writing – review and editing

### Author ORCIDs

Susan E Howlett http://orcid.org/0000-0001-5351-6308
Andrew D Rutenberg http://orcid.org/0000-0002-4264-6809

### Decision letter and Author response

Decision letter https://doi.org/10.7554/eLife.77632.sa1
Author response https://doi.org/10.7554/eLife.77632.sa2

## Additional files

### Supplementary files

• Supplementary file 1. Evidence for deficits that can reverse.
• Transparent reporting form

### Data availability

Source data files for all figures and summary statistics for all fitting parameters and diagnostics of the models are provided. Only pre-existing datasets were used in this study. Information about the datasets and data cleaning is in the methods section. Raw data for mouse dataset 3 are freely available from https://github.com/SinclairLab/frailty. Raw human data are available from https://www.elsa-project.ac.uk/accessing-elsa-data after registering. All code is available at https://github.com/Spencerfar/aging-damagerepair, (copy archived at swh:1:rev:4fe6f883d37dda6b9059c53aa9366f4ff2665a43). Our code for cleaning these raw datasets is provided.

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
