## [Editor Report]

The key contribution of this study is to evaluate the longitudinal change in frailty indices by tracking both accumulation of damage and repair of deficits (damage and repair transition rates), using a sophisticated mathematical modeling and a translational approach that spans mice and humans. A second key achievement of this study is to evaluate change in frailty indices and damage and repair transition in interventions that improve health in mice. Collectively this advances progress in translational geroscience by providing new insight regarding how we measure biological age that can aid assessment of aging-relevant interventions. The authors have provided extensive details that support the research frameworks presented in this report.

---

## [Decision Letter]

**Decision letter after peer review:**

Thank you for submitting your article "Measurements of damage and repair in aging mice and humans reveals that robustness and resilience decrease with age, operate over broad timescales, and are affected differently by interventions" for consideration by *eLife*. Your article has been reviewed by 4 peer reviewers, one of whom is a member of our Board of Reviewing Editors, and the evaluation has been overseen by Carlos Isales as the Senior Editor. The following individual involved in the review of your submission has agreed to reveal their identity: Konstantin Arbeev (Reviewer #3).

Essential revisions:

1. A key assumption is that an increase in FI is equivalent to a rise in damage, but this relationship is unknown and may ignore a number of important biological processes – including the dynamic interplay between both damage, compensatory processes/feedback, and protection/fitness. There is lingering concern that this may be over-conceptualized, and the true relationship between frailty index and damage is unknown. This should be addressed in the manuscript rationale. Language in discussion and conclusions should be tempered.

2. Additionally, 'repair' assumes a biological process to restore homeostasis, but in this manuscript refers to a switch from out-of-range to a return to in-range for variables and cut-points that are not defined in the manuscript. The issue of measurement error in individual components that switch from out-of-range to in-range or vice versa needs to be addressed. Variables and cutpoints must be detailed. Authors should also distinguish biological processes from mechanical definitions in rationale and discussion.

3. Some variables may represent irreversible processes of deterioration (e.g. loss of whiskers, kyphosis, palpable/visible tumor, etc.), while others may be more dynamic or have cutpoints with unknown validity for in vs out of range. It is anticipated that only a subset of those queried contributes to 'repair' while others irreversibly increase over time. This would challenge the validity of 'repair' models (and interpretation of Figure 5). As above, please include details on specific variables and their cutpoints. Please indicate which of these variables change dynamically over time – with evidence of both 'deficit' and 'repair' – versus those that are irreparable by definition or thresholds.

4. Additional conceptual issues are present in mouse models given typically shorter time scales and acute stress stimulus – in addition to the issues above. Most reviewers have requested additional details to support assumptions made in mouse models.

5. Detailed total follow-up time/time-course in both animal and human data is needed. For example, statistics on total follow-up times in human data. This is important for relative time courses across species and how many times deficits can be damaged/repaired within a given time or relative period of life.

6. Concern over the impact of selection bias/survivor bias in models (especially mice). Please address in the model and discussion.

7. Abstract requires clarity. Data, tests performed, and specific results should be presented as appropriate.

8. Please address specific recommendations and editorial suggestions made by each reviewer below.

*Reviewer #1 (Recommendations for the authors):*

I commend the authors for their efforts. The modeling of longitudinal data is an important area in aging that needs more attention. I also like the approach at a conceptual level and think the stratification of resilience vs robustness could be quite valuable. I appreciate the idea of making models that are generalizable and think the approach is most appropriate for the human data, but I think more attention needs to be paid to the technical and biological aspects of the variables being modeled for the mouse data.

General comments:

The mouse frailty index is a very useful tool for efficiently measuring the organismal state in large cohorts. A tradeoff for quickly measuring a broad range of health domains is that the individual measurements are low resolution and involve inherent subjectivity (which may be considered measurement error). It is very important to keep this in mind when using individual scores from the index. Some transitions are due to random measurement error and I believe this is especially likely with decreases (or 'resilience' transitions).

While I find the 'robustness' modeling quite interesting, I am skeptical as to how valid the 'resilience' modeling is. The first reason for this is that I do not believe many of the deficits in the mouse index are reversible under normal physiologic conditions. For example, it is exceptionally unlikely for a palpable/visible tumor to resolve in an aged mouse, thus any reversal that was observed must be due to random measurement error. I would conservatively estimate that the following components are not truly reversible in the contexts in which they were studied here: alopecia, loss of fur color, loss of whiskers, tumors, kyphosis, hearing loss, cataracts, corneal capacity, vision loss, rectal prolapse, genital prolapse.

When used as a sum of its components, the mouse frailty index is highly robust to these measurement errors as demonstrated many times in the literature. However, breaking apart the index into its components compromises this robustness, and I am not aware of literature which has validated the use of individual components of the index. How do the authors account for the measurement error which is inherent to the individual components of the mouse frailty index, and how might this error affect their models? I would be particularly concerned with how this would affect the models in figure 5.

Additionally, the authors state that "the observed damage is thought to occur due to natural stochastic transitions." I do not necessarily agree with that characterization for many of the pathologies measured here. For example, I would not consider vision loss a stochastic transition but rather a gradual degenerative process. Stochasticity involves when it crosses a threshold that can be measured/detected by the researcher. The authors state the following assumption: "We assume these time intervals are small so that we use constant rates within each time-interval to approximate these integrals." Small is subjective and the time intervals are up to 2 months which is 7-10% of a mouse lifespan. Can the authors justify this assumption and/or discuss the implications of such an assumption to their models? The time scale should also be considered in the context of the specific measurements being taken.

My final high-level comment regards selection bias. As these studies progress and individuals die, the study population drastically changes. How do the authors' models account for this selection bias? Can the results of the modelling tell us anything about the nature of the populations as selection occurs? And finally, would there be value in performing this modeling as a function of "biological age" (ie. time to death) versus chronological age?

Specific comments:

The authors should specify which packages and versions were used for the development of their models. When I ran some of the scripts from Github they threw pandas warnings, leading me to suspect that I am not using the development versions. This is important if the authors expect future use of their models. I would encourage the authors to include package information both in the manuscript and on the Github.

I would recommend the authors be specific when describing their results. Many complex models and statistical tests are generated and it sometimes is unclear what is being referred to. For example, in the section "The acceleration of damage accumulation…" the statement "In mice, this is seen in every dataset and is significant at the indicated ages" is unclear what statistical test is being referred to. I would also recommend the authors be more precise in their result descriptions. For example, in the section "Interventions modify damage…" the statement "This curvature is strongly reduced…" is subjective and would be better supported by a description of specific statistical results.

The authors should be careful with the language used regarding wealth in the human dataset. Wealth is not an intervention. And wealth per se does not influence frailty as worded in methods. In these data wealth is a variable which is often associated with health outcomes including frailty.

In discussion: "…where female live longer but have higher FI scores than males." Female humans live longer but female B6 mice usually live shorter.

The citation the authors use for ELSA points to the database. PMID 23143611 also needs to be cited where the study and data collection procedures are described. It also seems like an omission to not cite PMID 31665163 which explores the relationship between various factors, including wealth, and frailty in the same dataset. I would also recommend the authors include a supplemental table with the specific mouse measurements and human questionnaire items that comprise the respective frailty indices.

The authors should provide more detail on the human frailty variables in the methods. I am assuming the binary responses to questions HEADLA and HEADLB were used, but this should be clarified/explicitly stated in the methods.

Comments on mouse pre-processing methods: In the mouse frailty index, 29 out of 31 parameters are measured on a [0, 0.5, 1] scale, with only weight and temperature using [0, 0.25, 0.5, 0.75, 1] scale, and it looks to me like the Schultz dataset still used [0, 0.5, 1] for these variables. So I don't find the statement "a single repair or damage transition can be interpreted as taking a step of size 0.25 on the fractional deficit scale" to be accurate because 29 out of 31 parameters only have the resolution to move in 0.5 unit increments.

Page 5: Authors state that "in each of the datasets, there is a strong decrease in repair rates and increase in damage rates with age (except in mouse dataset 2 for damage rates)." This doesn't seem to be the case for repair rates in humans entering ELSA study at age 50-70 (CI for p cross 0).

I am also somewhat unclear about the values for the rates in figure 2. In figure S2-1 the authors show that the model fits the data reasonably well. However, when looking at the rates in figure 2, the repair rates appear to be consistently higher than the damage rates. For example, in the Keller dataset males at 22 months, as frailty is consistently increasing, the modeled repair rate appears to be nearly double the modelled damage rate. Overall, the comparative magnitudes of the rates don't seem consistent with the mouse data, where there it is relatively rare to see a "population level" reduction in frailty at a subsequent timepoint (tends to stay the same or increase over time as shown in Figure 2S-2). Same comment for figure 4S2 human data.

Figure S2-1. It is hard to see the bars of the histogram in comparison to the dots. I would recommend making the dots smaller. Quantifying and plotting the deviation of observed vs posterior samples may also make interpretation easier.

The repair rate in the human 70-80 data (2d) appears to have a negative curvature, but this doesn't seem to be reflected in the second derivative plot (3e, teal line centered on zero).

I find the models in figure 5 somewhat difficult to interpret considering the extensive censoring that is occurring. Could the authors attempt a more 'plain language' interpretation of these curves considering the censoring? It seems like the interpretation would be that at 3 months after damage there is only a 75% probability of that damage being repaired? Also, could the authors interpret the drastic reduction in probability occurring in both damage and repair of the Schultz data at ~3 months?

*Reviewer #3 (Recommendations for the authors):*

I find the result on sex differences in effects of interventions on robustness and resilience very interesting so consider mentioning this in the abstract.

I would like to see more discussion on why deficits can be repaired, especially for some deficits which may seem to represent irreversible processes of deterioration. One example that is not trivial for me to understand (as I am not a specialist in mice) is the repair of the deficit "loss of whiskers." Do they just grow up again after a while? If they do, what constitutes the event of repair (just when they start growing again or when they are fully grown, assuming there is a measure for this)?

Some statistics on the total follow-up times in human data would be helpful. It can help put the estimates of repair/damage scales in the context to provide insight on how many times the deficit can potentially be damaged/repaired in the time period of data collection.

As you indicated in Methods, "The individuals selected from ELSA with wealth data do not have mortality data available, simplifying the model from the joint model used above for mice." In Results, you wrote, "This is a joint longitudinal-survival model, which couples the damage and repair rates together with mortality." This narrative should be changed then to reflect the fact that the joint model was only used in mice data.

Some readers may not be familiar with Bayesian methods so it would help provide in Methods some discussion on the selection of priors – is this a convenience/traditional choice or some other reasons? Also, some narrative on the sensitivity of results to a different choice of priors would be helpful, in my view.

---

## [Author Response]

Essential revisions:1. A key assumption is that an increase in FI is equivalent to a rise in damage, but this relationship is unknown and may ignore a number of important biological processes – including the dynamic interplay between both damage, compensatory processes/feedback, and protection/fitness. There is lingering concern that this may be over-conceptualized, and the true relationship between frailty index and damage is unknown. This should be addressed in the manuscript rationale. Language in discussion and conclusions should be tempered.

The frailty index (FI) is a summary health measure that is useful to predict (and correlates with) many adverse outcomes. It is a percentage of individual health attributes that are considered “damaged”. Individual binarized attributes are not absolute: “damaged” (or “deficit”) attributes are not entirely damaged, while “healthy” attributes are not entirely healthy. Nevertheless, we are considering transitions between these binary variables, so we use the language of damage and repair. Since the FI is the average of the binary attributes, we use the same language for changes of the FI to distinguish increases of FI (damage) from decreases (repair).

While the FI is a summary measure of health it is not the only one, and different summary measures will provide different natural measures of damage and repair. We have tempered our language throughout to make it more clear what we have done (with binary variables) and what questions this raises (with the connection to non-binary approaches and conceptualizing damage). In particular we ignore nonlinearities (compensatory processes, feedback, protection) and do not assess overall fitness apart from the FI. We have also added a caveat paragraph in the discussion (p.6 lines 251-258). See also penultimate paragraph in the introduction (p.2 lines 56-64).

2. Additionally, 'repair' assumes a biological process to restore homeostasis, but in this manuscript refers to a switch from out-of-range to a return to in-range for variables and cut-points that are not defined in the manuscript. The issue of measurement error in individual components that switch from out-of-range to in-range or vice versa needs to be addressed. Variables and cutpoints must be detailed. Authors should also distinguish biological processes from mechanical definitions in rationale and discussion.

As mentioned above, we define “repair” to be a binary transition from deficit (1) to healthy (0) state. We report aggregate repair or damage rates of individual binarized attributes. Whether these correspond to organismal repair or damage conceptualized differently is an interesting question that we do not address. Given the proven utility of the FI to predict (or associate with) adverse outcomes, we do expect that aggregate repair (resilience) of individual binarized attributes is associated with (but not the same as) improved health at the population level as assessed by other health measures – and similarly with damage.

To be clear, we start with binarized variables (a mechanical definition) and analyze aging in that context. We obtain results that capture biological processes of aging, but through the lens of the binarized variables. We are not asserting that the biological processes are binary, but simply that binary variables can be used to study aging. Given that much of medicine uses binary variables (in diagnosis and treatment), and that much aging data is only binary (from e.g. self-report data) we feel that this is a useful and promising approach. We have clarified these points in the text (see e.g. p.2 lines 56-64 and p.6 lines 251-258). In the methods (p.8 and 9), we have also included more details about the mouse and human deficits used.

Transitions of the binarized variables could correspond to small changes of underlying continuous processes, which also raises the question of measurement error resulting from small changes in the underlying biological state. While we cannot avoid measurement error, we do not feel that it causes our key results. We have two reasons. The first is that we do not expect age-dependence in measurement error, so we do not feel that the age-dependence we observe for damage and repair are due to measurement error. Second, we have performed a sensitivity analysis by removing potentially erroneous damage and repair events. For adjoining and isolated damage and repair events characterized by a sequence of variable states {0,0,1,0,0} or {1,1,0,1,1} (i.e. isolated states surrounded by at least two previous and two subsequent of the opposite state), we remove the damage/repair event resulting in states {0,0,0,0,0} or {1,1,1,1,1}. With these isolated events removed, we see little change in the time-dependence of the repair and damage rates vs age and the time-scales of resilience and robustness when compared to our original results. We have added Figure 5 —figure supplements 3 and 4 and Section 5 in the methods describing this approach. We have also added discussion of measurement errors in the discussion (p.6 lines 259-268). We also note that measurement error for mice has been assessed previously, and is small (Feridooni et al., 2015; Kane et al., 2017).

3. Some variables may represent irreversible processes of deterioration (e.g. loss of whiskers, kyphosis, palpable/visible tumor, etc.), while others may be more dynamic or have cutpoints with unknown validity for in vs out of range. It is anticipated that only a subset of those queried contributes to 'repair' while others irreversibly increase over time. This would challenge the validity of 'repair' models (and interpretation of Figure 5). As above, please include details on specific variables and their cutpoints. Please indicate which of these variables change dynamically over time – with evidence of both 'deficit' and 'repair' – versus those that are irreparable by definition or thresholds.

The reviewers have pointed out that some of the variables measured in the mouse frailty index may be “irreversible” processes and thus not susceptible to any extrinsic or intrinsic repair processes. This is a good question that we have addressed in our revised manuscript. We have considered all the deficits in the mouse frailty index to determine whether there is evidence that they can repair, either spontaneously through intrinsic repair processes or extrinsically by interventions such as drug treatment or lifestyle changes (e.g. exercise, caloric restriction, drug treatment etc). Interestingly, we found that virtually all the deficits in the frailty index can potentially reverse, especially in response to an intervention. The only deficits that may not reverse on their own or in response to interventions are microphthalmia (but this is an active area of drug discovery) and vaginal/uterine or penile prolapses (but these can be corrected surgically). We have created Supplementary File 1 (with references) to support the concept that almost all deficits in the murine frailty index can repair either on their own or in response to interventions, including many interventions that are used in aging research. We have added a new paragraph that discusses this to our revised manuscript (p.3 lines 99-105). We have also listed all health attributes used in this study in the methods (p.8 and 9).

Despite ubiquitous repair, not all deficits repair equally. We have included Figure 5 —figure supplement 4 showing the repair counts for each deficit in the 3 mouse datasets. In particular, we find deficits like “Cataracts”, “Tumours”, “diarrhea”, and “vaginal/uterine/penile prolapses” are rarely repaired. We also show which damage/repair events are removed in the sensitivity analysis of isolated damage/repair events. This shows that some of these deficits that are rarely repaired may also be the result of measurement error, but as above, this small amount error does not effect the results due to the rarity of repair for these deficits.

A broader question is how our results depend on the underlying binarization thresholds of individual variables. We have not explored that question here (and cannot for the majority of our variables that are only measured as binary or ordinal values), rather we have used the binarized attributes as previously published. While we do not expect that they will drastically change our aggregate results, we do expect that detailed transition rates of individual health attributes will depend on the binarization thresholds. Previous work on binarization thresholds of the FI has shown weak dependence of predictive power on small changes of the thresholds (Stubbings, G., Farrell, S., Mitnitski, A., Rockwood, K. and Rutenberg, A. Informative frailty indices from binarized biomarkers. *Biogerontology* 21, 345–355 (2020)). This is an interesting question for future study in the context of robustness and resilience.

4. Additional conceptual issues are present in mouse models given typically shorter time scales and acute stress stimulus – in addition to the issues above. Most reviewers have requested additional details to support assumptions made in mouse models.

As discussed in the paper (p.6 lines 239-250) we do not use acute stress stimuli, and our timescales are limited to the available data (p.6 lines 251-258).

5. Detailed total follow-up time/time-course in both animal and human data is needed. For example, statistics on total follow-up times in human data. This is important for relative time courses across species and how many times deficits can be damaged/repaired within a given time or relative period of life.

We have included follow-up times (p. 8 and 9).

6. Concern over the impact of selection bias/survivor bias in models (especially mice). Please address in the model and discussion.

The mouse study was performed without replacement and studied with a joint-model to capture survivor effects. The human study contained dropout and replacement, but without knowledge of mortality (so the joint survival model was not used for humans). We did not model drop-out effects in the human study. We have added a paragraph on potential population biases in the discussion (p.6 lines 259-268).

7. Abstract requires clarity. Data, tests performed, and specific results should be presented as appropriate.

We have clarified our abstract.

8. Please address specific recommendations and editorial suggestions made by each reviewer below.

See below.

Reviewer #1 (Recommendations for the authors):I commend the authors for their efforts. The modeling of longitudinal data is an important area in aging that needs more attention. I also like the approach at a conceptual level and think the stratification of resilience vs robustness could be quite valuable. I appreciate the idea of making models that are generalizable and think the approach is most appropriate for the human data, but I think more attention needs to be paid to the technical and biological aspects of the variables being modeled for the mouse data.General comments:The mouse frailty index is a very useful tool for efficiently measuring the organismal state in large cohorts. A tradeoff for quickly measuring a broad range of health domains is that the individual measurements are low resolution and involve inherent subjectivity (which may be considered measurement error). It is very important to keep this in mind when using individual scores from the index. Some transitions are due to random measurement error and I believe this is especially likely with decreases (or 'resilience' transitions).While I find the 'robustness' modeling quite interesting, I am skeptical as to how valid the 'resilience' modeling is. The first reason for this is that I do not believe many of the deficits in the mouse index are reversible under normal physiologic conditions. For example, it is exceptionally unlikely for a palpable/visible tumor to resolve in an aged mouse, thus any reversal that was observed must be due to random measurement error. I would conservatively estimate that the following components are not truly reversible in the contexts in which they were studied here: alopecia, loss of fur color, loss of whiskers, tumors, kyphosis, hearing loss, cataracts, corneal capacity, vision loss, rectal prolapse, genital prolapse.

All mouse deficits are demonstrably reversible (see our new Supplementary File 1).

When used as a sum of its components, the mouse frailty index is highly robust to these measurement errors as demonstrated many times in the literature. However, breaking apart the index into its components compromises this robustness, and I am not aware of literature which has validated the use of individual components of the index. How do the authors account for the measurement error which is inherent to the individual components of the mouse frailty index, and how might this error affect their models? I would be particularly concerned with how this would affect the models in figure 5.

Measurement error is an understudied effect in aging research; it is not typically assessed or reported. We expect that measurement error would be a source of “white noise”, that would contribute to both damage and repair at the same rate. Effects of age or intervention should not be affected by such measurement errors, and these effects make up our results. We have checked the sensitivity of the results of Figure 5 to exclude (“fix”) damage or repair events that occur in isolation away from the first or last element in a time-series (so “11011” would be fixed to “11111”). The results are shown in Figure 5 —figure supplements 3 and 4, and do not affect our qualitative conclusions. We also note that measurement error for mice has been assessed previously, and is small (Feridooni et al., 2015; Kane et al., 2017).

Additionally, the authors state that "the observed damage is thought to occur due to natural stochastic transitions." I do not necessarily agree with that characterization for many of the pathologies measured here. For example, I would not consider vision loss a stochastic transition but rather a gradual degenerative process. Stochasticity involves when it crosses a threshold that can be measured/detected by the researcher.

We have improved the wording. We were contrasting natural processes vs applied stressors.

The authors state the following assumption: "We assume these time intervals are small so that we use constant rates within each time-interval to approximate these integrals." Small is subjective and the time intervals are up to 2 months which is 7-10% of a mouse lifespan. Can the authors justify this assumption and/or discuss the implications of such an assumption to their models? The time scale should also be considered in the context of the specific measurements being taken.

This approximation is implemented for computational convenience so that we can use Poisson statistics despite having non-zero timesteps. While this approximation is common in the modelling literature, it is uncontrolled. Systematic effects of this approximation do not appear to be large, however, since the average model damage and repair rates (lines in Figure 2) are close to the measured rates (points in Figure 2) – and the individual distributions (Figure 2 supplement 1) are close too. Furthermore we reran our mouse analysis using rates evaluated at endpoints rather than the start of time intervals (methods equation 3 and 4 with λ(t_j+1_) rather than λ(t_j_)). We see no strong effects (compare with Figure 4).

My final high-level comment regards selection bias. As these studies progress and individuals die, the study population drastically changes. How do the authors' models account for this selection bias? Can the results of the modelling tell us anything about the nature of the populations as selection occurs? And finally, would there be value in performing this modeling as a function of "biological age" (ie. time to death) versus chronological age?

For the mouse studies there was no replacement after mortality. Our analysis was with a joint model that modelled both health and mortality, so any survivor effect is not a bias but rather is the result of aging. To mitigate selection bias in the human data, we treated onset ages distinctly. There are nevertheless well known “healthy volunteer” effects that would bias the original population that we did not consider. Furthermore, we did not have human mortality data so we could not treat survivor bias effects. Studying the selection effects themselves (due to mortality or due to selection bias) would be interesting, but beyond the scope of this study. Modelling with respect to time to death (rather than chronological age) would also be interesting. Such a study would be limited to individuals who die, and so would exclude all of the human data we used – where we had no mortality information.

Specific comments:The authors should specify which packages and versions were used for the development of their models. When I ran some of the scripts from Github they threw pandas warnings, leading me to suspect that I am not using the development versions. This is important if the authors expect future use of their models. I would encourage the authors to include package information both in the manuscript and on the Github.

The Github includes package information. This has also been added to the manuscript (p.14 lines541-544). A version update in pandas resulted in warnings. We have fixed these for the latest version of pandas.

I would recommend the authors be specific when describing their results. Many complex models and statistical tests are generated and it sometimes is unclear what is being referred to. For example, in the section "The acceleration of damage accumulation…" the statement "In mice, this is seen in every dataset and is significant at the indicated ages" is unclear what statistical test is being referred to. I would also recommend the authors be more precise in their result descriptions. For example, in the section "Interventions modify damage…" the statement "This curvature is strongly reduced…" is subjective and would be better supported by a description of specific statistical results.

We have described in the text and figure captions the statistical tests done. The first example ("seen in every dataset") is addressed p.4 lines 148-151. The second example ("strongly reduced") is quantitatively addressed in Figure 3 —figure supplement 1. We use strongly here to summarize that the credible intervals of exercise effect (in panel e) exclude zero effect.

The authors should be careful with the language used regarding wealth in the human dataset. Wealth is not an intervention. And wealth per se does not influence frailty as worded in methods. In these data wealth is a variable which is often associated with health outcomes including frailty.

We agree; we have cleared up language regarding wealth to emphasize it is not an intervention.

In discussion: "…where female live longer but have higher FI scores than males." Female humans live longer but female B6 mice usually live shorter.

We have fixed this; in our results we also observe these sex effects in the FI.

The citation the authors use for ELSA points to the database. PMID 23143611 also needs to be cited where the study and data collection procedures are described. It also seems like an omission to not cite PMID 31665163 which explores the relationship between various factors, including wealth, and frailty in the same dataset. I would also recommend the authors include a supplemental table with the specific mouse measurements and human questionnaire items that comprise the respective frailty indices.

We have added the two references, and have listed all of the mouse and human items in the methods.

The authors should provide more detail on the human frailty variables in the methods. I am assuming the binary responses to questions HEADLA and HEADLB were used, but this should be clarified/explicitly stated in the methods.

All variables are now listed in the methods. All of our human variables are binary (with values 0 or 1).

Comments on mouse pre-processing methods: In the mouse frailty index, 29 out of 31 parameters are measured on a [0, 0.5, 1] scale, with only weight and temperature using [0, 0.25, 0.5, 0.75, 1] scale, and it looks to me like the Schultz dataset still used [0, 0.5, 1] for these variables. So I don't find the statement "a single repair or damage transition can be interpreted as taking a step of size 0.25 on the fractional deficit scale" to be accurate because 29 out of 31 parameters only have the resolution to move in 0.5 unit increments.

We have clarified this section (p.8-9 lines 367-379). To keep the FI values comparable between studies, we have used the same [0, 0.25, 0.5, 0.75, 1] scale for all health attributes, with a step of size 0.25. This means that for attributes with values [0, 0.5, 1], the only observed steps will be of size 2. For human data, all attributes are binarized and on the same [0, 1] scale with steps of size 1.

Page 5: Authors state that "in each of the datasets, there is a strong decrease in repair rates and increase in damage rates with age (except in mouse dataset 2 for damage rates)." This doesn't seem to be the case for repair rates in humans entering ELSA study at age 50-70 (CI for p cross 0).

The points are binned averages, which are useful as guides to the eye. The lines are from our model. The error bars on the points (and the variability of the lines) indicates uncertainties. As humans enter the ELSA study at ages 50 and 70 there are large uncertainties.

I am also somewhat unclear about the values for the rates in figure 2. In figure S2-1 the authors show that the model fits the data reasonably well. However, when looking at the rates in figure 2, the repair rates appear to be consistently higher than the damage rates. For example, in the Keller dataset males at 22 months, as frailty is consistently increasing, the modeled repair rate appears to be nearly double the modelled damage rate. Overall, the comparative magnitudes of the rates don't seem consistent with the mouse data, where there it is relatively rare to see a "population level" reduction in frailty at a subsequent timepoint (tends to stay the same or increase over time as shown in Figure 2S-2). Same comment for figure 4S2 human data.

Our repair/damage rates are calculated per the number of variables available to damage/repair. As such, repair rates can be above damage rates (per variable) even when the total damage rate (summed over all variables) exceeds the total repair rate. This can occur when most variables are undamaged. We have emphasized this at the end of the first section of the results.

Figure S2-1. It is hard to see the bars of the histogram in comparison to the dots. I would recommend making the dots smaller. Quantifying and plotting the deviation of observed vs posterior samples may also make interpretation easier.

Done.

The repair rate in the human 70-80 data (2d) appears to have a negative curvature, but this doesn't seem to be reflected in the second derivative plot (3e, teal line centered on zero).

The second derivatives of the human age 70-80 repair rate are small and are consistent with zero. The lines in Figure 2d should be compared with the lines in Figure 3e.

I find the models in figure 5 somewhat difficult to interpret considering the extensive censoring that is occurring. Could the authors attempt a more 'plain language' interpretation of these curves considering the censoring? It seems like the interpretation would be that at 3 months after damage there is only a 75% probability of that damage being repaired? Also, could the authors interpret the drastic reduction in probability occurring in both damage and repair of the Schultz data at ~3 months?

(See also detailed point 22 from reviewer #3) While we have tried to correct for interval censoring, the available methods only makes the survival curves self-consistent – it does not impose a model for short measurement intervals that are not observed. The sharp drop of the survival curves reflect the absence of measurements at shorter measurement intervals. As such, the survival curves after the sharp drops are more reliable – but also comparisons in Figure 5 between intervention and control, between male and female, or in Figure 5 -- supplemental Figures1 and 2 between different health attributes.

We have labeled the axes with the plain language interpretation. Yes, for mice most damaged deficits (approx. 75%) are repaired by 3 months.

Reviewer #3 (Recommendations for the authors):I find the result on sex differences in effects of interventions on robustness and resilience very interesting so consider mentioning this in the abstract.

Done.

I would like to see more discussion on why deficits can be repaired, especially for some deficits which may seem to represent irreversible processes of deterioration. One example that is not trivial for me to understand (as I am not a specialist in mice) is the repair of the deficit "loss of whiskers." Do they just grow up again after a while? If they do, what constitutes the event of repair (just when they start growing again or when they are fully grown, assuming there is a measure for this)?

See new Supplementary file 1 on repairability of mouse deficits. All deficits are evaluated with respect to standard criteria. For example, for whiskers the score is 0 if all whiskers are present, 0.5 if a reduced number are present, and 1 if no whiskers are present.

Some statistics on the total follow-up times in human data would be helpful. It can help put the estimates of repair/damage scales in the context to provide insight on how many times the deficit can potentially be damaged/repaired in the time period of data collection.

We have added follow-up information in the methods (p.9 in Human data and pre-processing).

As you indicated in Methods, "The individuals selected from ELSA with wealth data do not have mortality data available, simplifying the model from the joint model used above for mice." In Results, you wrote, "This is a joint longitudinal-survival model, which couples the damage and repair rates together with mortality." This narrative should be changed then to reflect the fact that the joint model was only used in mice data.

Fixed.

Some readers may not be familiar with Bayesian methods so it would help provide in Methods some discussion on the selection of priors – is this a convenience/traditional choice or some other reasons? Also, some narrative on the sensitivity of results to a different choice of priors would be helpful, in my view.

We have added discussion in the methods (p.11 lines 458-468) describing the choice of prior. Our choice of weak (or “noninformative”) priors are a matter of computational convenience. We have chosen standard weak priors. Changing weak priors does affect the quantitative results, but weakly, and does not change the qualitative results. Any model or analysis pipeline has many such assumptions and hyperparameters; changing any of them will have similar effects. One can think of choice of weak priors as a choice of hyperparameter.